# A Building Project-Based Industrialized Construction Maturity Model Involving Organizational Enablers: A Multi-Case Study in China

**Guangbin Wang [1], Huan Liu [1],\*, Heng Li [2], Xiaochun Luo [2] and Jiaxi Liu [1]**

[1]  Department of Construction Management and Real Estate, School of Economics and Management, Tongji University, 1239 Siping Road, Shanghai 200092, China; gb_wang@tongji.edu.cn (G.W.); victor-liu@outlook.com (J.L.)

[2]  Department of Building and Real Estate, Hong Kong Polytechnic University, Room ZN1002, Hung Hom, Kowloon, Hong Kong SAR, China; heng.li@polyu.edu.hk (H.L.); bsericlo@polyu.edu.hk (X.L.)

\*   Correspondence: huanliu2017@tongji.edu.cn

**Abstract:** Industrialized construction (IC) as a promising construction mode has been increasingly adopted in China due to its advantages of enhancing productivity and reducing the labor intensiveness in the construction industry. An objective and systematic evaluation of the IC mode is essential by clarifying the current weak areas in application and improving project performance. The meager existing studies have considered evaluating the IC maturity of prefabricated construction projects from the perspective of project governance. This study proposed an industrialized construction maturity model (ICMM) involving organizational enablers by employing the framework of the well-established European Foundation for Quality Management (EFQM) excellence model. The evaluation indicator system consisting of two parts (i.e., "enablers" and "results") were abstracted by a literature review and expert interviews. The analytic hierarchy process (AHP) approach was used to weight the indicators. The maturity of IC projects was rated as four levels (i.e., initial, upgraded, integrated, and optimal levels). The proposed ICMM was validated by conducting a multi-case study, four typical building projects that adopted prefabricated construction techniques in Shanghai were selected and evaluated by the proposed ICMM. Results showed that the ICMM can objectively and comprehensively realize the status quo of IC projects and help managers to identify weak areas of the current IC projects and performance improvement paths from the perspective of project governance. The ICMM was also evaluated to demonstrate its applicability and reliability through expert interviews.

**Keywords:** maturity model; project performance evaluation; industrialized construction; performance improvement; organizational enablers; CMM; the EFQM excellence model

## 1. Introduction

Industrialized construction (IC), as a promising construction mode that can enhance production efficiency and reduce the labor intensiveness in the construction industry [1–5], has played an important role in facilitating the sustainability performance of construction projects recently [6]. Different terms exist for describing IC mode from the perspective of technology system adoption, such as "module", "prefabrication", "offsite prefabrication", "modularization", and "preassembly" [7]. In this paper, IC is defined as a form of modern manufacturing, transportation, installation, and scientific management of construction, which differs from the decentralized, low-level, and low-efficiency handicraft production in the traditional construction industry [8–11]. Considering the advantages of the IC mode, cities, such as Beijing, Shanghai, and Shenzhen, in China have carried out a series of IC technology experiments on prefabricated construction building projects and implemented preferential policies and incentives

for builders. However, the desired efficiency (e.g., cost savings and duration shortening) has not yet been achieved. Such problems related to market demand, production networks, technology docking, and supply consolidation have emerged in the low productivity of large-scale onsite construction and prevented the progress of the industrialization of construction [12].

To clarify and address the above problems of IC implementation, recognizing the developing status of IC application in real-life building projects by clarifying those weak areas and exploring the performance improvement strategies could be the potential solution. In practice, some published standards/criteria in different levels (e.g., the Standard for Assessment of Prefabricated building—GB/T51129-2017, and the Evaluation Standards for Industrialized Housing in Shanghai—DG/TJ08-2198-2016) have mainly been adopted to assess the performance of the construction process and the results involving entities. Additionally, academia has also established maturity evaluation models of IC from different perspectives to reveal the implementing status and explore the development progress based on the capability maturity model (CMM) theory [13]. These existing evaluation metrics of IC have presented the weak areas but have failed to indicate the roadmap of further improvement, especially from the perspective of project governance.

In response to the above limitations, this study provided an adaptive IC maturity evaluation model to reveal the weakness of the IC implementing process and confirm the roadmap for performance improvement, namely, the industrialized construction maturity model (ICMM). By employing the concept of CMM in project management [14], the IC maturity can be defined as the project organization's capability to successfully and reliably achieve predetermined project goals by adopting IC technology and a corresponding management approach. The ICMM that need to achieve an improved IC maturity from perspective of project governance was established by integrating the CMM theory [15] and the framework of the EFQM excellence model [16,17]. The CMM as a typical evaluation model can imply potential for growth in capability [15], and the European Foundation for Quality Management (EFQM) excellence model considers results (what needs to be improved) and enablers (how organization to do to improve it) [16,17]. All the indicators were abstracted through a comprehensive literature review and expert interviews. The analytic hierarchy process (AHP) was employed to determine the weights. The final model included the "enablers" and "results" perspectives with four maturity rating settings (i.e., initial, upgraded, integrated, and optimal levels). A multi-case study by evaluating four typical building projects that adopted the prefabricated construction technology in Shanghai was implemented to validate the proposed ICMM. The results show that the ICMM with high applicability and reliability was validated, the proposed ICMM can comprehensively reveal the status quo of IC mode implementation and help managers to identify weak areas of the current IC projects and take practical improvement measures.

The rest of this study is organized as follows. Section 2 summarizes the existing literature on IC maturity evaluation, maturity evaluation models, and the application of the EFQM excellence model in the construction industry. Section 3 outlines the research methods. The establishment of the industrialized construction maturity model (ICMM) is described in Section 4. Section 5 presents a multi-case study that was conducted to validate the ICMM. Section 6 discusses the general maturity of IC projects in China and demonstrates the applicability and reliability of the ICMM. Section 7 presents conclusions by highlighting the main contributions and limitations of the study.

## 2. Literature Review

Developing the ICMM is a comprehensive original work, of which the theoretical framework integrates the CMM theory and the EFQM excellence model. In this section, we first review the related work on methods or models of evaluating industrialized construction projects; second, the related work of the maturity theory for construction projects from previous studies and the introduction of the CMM theory is given; next, the concept of the EFQM excellence model with its applications in the construction industry are introduced; and finally, a summary with the motivation of this study is identified.

### 2.1. Methods or Models of Evaluating Industrialized Construction Projects

Previous studies on construction performance evaluation have indicated the recent trend on performance improvement going from the management level to the governance level. Traditionally, construction projects mainly involve three objectives (i.e., cost, time, and quality), namely the "iron triangle" indicates three standards in project success [18]. Subsequently, more diversified evaluation methods for project performance have been proposed. From the corporative perspective, key performance indicators (KPIs) [19] and the capability maturity model (CMM) [15] are most commonly used. From the perspective of business, the balanced scorecard card (BSC) model [20] and the EFQM excellence model were adopted. Table 1 presents some existing theories of project performance assessment that have been involved in the current construction industry. The indicators of construction project performance assessment have gone through the process from single, static, and stage-based, to multi-dimensional, dynamic, and life cycle gradually, which has indicated the recent trend on performance improvement going from the management level to the governance level.

**Table 1.** Project performance assessment theory.

| Assessment Theory | | Evaluation Dimensions | Attributes | Ref. |
|---|---|---|---|---|
| Traditional Theory | Financial Evaluation | Invest Return Rate, Cost-Effective Ratio, etc. | Static, Single Dimensional, Stage | [21] |
| | Iron Triangle | Cost, Quality, Schedule | Static, Multi-Dimensional, Stage | [18] |
| Mordent Theory | Balanced Scorecard Card | Finance, Customer, Internal Processes, Innovative Learning | Dynamic, Multi-Dimensional, Stage | [20] |
| | Key Performance Indicators (KPIs) | Finance, Operations, Organization | Dynamic, Multi-Dimensional | [19] |
| | Criteria for Performance Excellence | Leadership, Strategy, Customer and Market Measurement, Analytics and Knowledge Management, Human Resources, Process Management, Business Results | Dynamic, Multi-Dimensional | [22] |
| | Maturity Model | Capability Maturity Level | Dynamic, Multi-Dimensional | [17] |

Meager existing studies have attempted to explore IC maturity evaluation. Some trails on recognizing the IC mode and status assessment of IC applications have been made. Hong et al. [23] and Li et al. [24] explored barriers and critical success factors for facilitating IC application in China. Liu et al. [13] proposed a supplier management evaluation criteria system for prefabricated construction projects from five dimensions, namely, the procurement process, operation efficiency, relationship coordination, strategy alignment, and corporate social responsibility. A maturity grid with five levels was designed to present a continuous improvement in supplier management. Jerker et al. [25] described the development of industrialized house building (IHB) to increase the understanding of the field, which provided an orientation for the leading companies to structure and organize their work within industrialization and gave valuable advice to practitioners with interest in the field. In summary, the existing studies on the evaluation of IC projects have only revealed where the weakness lies in the process and outcome but failed to provide effective suggestions of performance improvement for the IC project organization, and the existing indicators in the evaluation system barely involved the organizational enablers. It is essential to integrate barriers and critical success factors, a maturity model framework, and the organizational enablers theory to bridge the research gap of the IC maturity evaluation method. Additionally, IC maturity can be defined as "the project organization's capability to successfully achieve predetermined project goals by adopting IC technology and corresponding management approach".

## 2.2. Maturity Models and Maturity Evaluation in Construction Management

The maturity model originated from the software manufacturing industry, and was developed by the Carnegie Mellon University Software Engineering Institute under the leadership of Watts Humphrey, namely the capability maturity model for software (CMM or SW-CMM). The CMM is organized into five maturity levels (i.e., initial, repeatable, defined, managed, and optimizing). Except for Level 1, each maturity level is decomposed into several key process areas that indicate the areas an organization should focus on to improve its software process [26]. The rating components of the CMM, to assess an organization's process maturity, are its maturity levels, key process areas, and their goals. Each key process area is further described by informative components: key practices, sub practices, and examples. The key practices describe the infrastructure and activities that contribute most to the effective implementation and institutionalization of the key process area.

The CMM has been used as a fundamental and popular tool for measuring project management performance [14,15,27]. Various maturity models from different fields have been proposed, and Table 2 presents the common maturity models with their basic information. Most of the common maturity models were proposed based on the CMM by academia from the USA and UK, and both quantitative analysis and qualitative analysis were adopted to implement the maturity evaluation. The maturity of project management (e.g., the project management maturity model (PMMM) proposed by the office of government commerce, and the organizational project management maturity model (OPM3) proposed by the project management institute) is the most common maturity area [14].

**Table 2.** Common maturity models with their characteristics.

| Maturity Model | Number of Levels | Application Area | CMM-Based | Analysis Method | |
|---|---|---|---|---|---|
| | | | | Quantitative | Qualitative |
| Capability Maturity Model Integrated (*CMMI*) [28] | 5 | Software industry | √ | √ | √ |
| Construction Industry Macro Maturity Model (*CIM3*) [29] | 4 | Construction industry | √ | √ | √ |
| Organizational Project Management Maturity Model (*OPM3*) [30] | 3 | Project management | √ | √ | |
| Berkley Project Management Process Maturity Model (*Berkley PM2*) [28] | 4 | Project management | √ | √ | |
| Portfolio, Programme and Project Management Maturity Model (*OPM3*) [30] | 5 | Project management | √ | √ | √ |
| Standardized Process Improvement for Construction Enterprises (*SPICE*) [31] | 5 | Construction industry | √ | √ | √ |
| Change Management Maturity Model (*CM3*) [28] | 5 | Construction industry | √ | √ | √ |
| Maturity Assessment Grid from the Strategic Forum for Construction (*MAG*) [28] | 5 | Construction industry | | √ | |
| Project Management Maturity Model (*PMS-PMMM*) [30] | 5 | Project management | √ | √ | √ |
| Kerzner Project Management Maturity Model (*K-PMMM*) [28] | 5 | Project management | √ | √ | √ |

In the construction industry, different maturity models were constructed to facilitate various project management tasks involving BIM application [32], knowledge management [33,34], project risk management [35–38], construction supply chain management [13,39], safety of construction contractors [40], information management [41,42], and detailed engineering maturity [43]. Considering

the project as an integral whole, previous studies have also explored the maturity level of the program management organization maturity [44], project management in different industries [45], and growth management in the developing construction industry [46]. All existing evaluation methods for construction management proved that the maturity model can reveal weak areas and indicate the growth steps of the performance that help managers to reach project objectives better, as well as the CMM theory as the common base has been used widely in the construction industry.

### 2.3. The EFQM Excellence Model Application in the Construction Industry

The EFQM excellence model, which is most widely used in Europe, has provided organizations with a tool for self-business evaluation and improvement [16,17]. The EFQM excellence model consists of the "enablers" area with five dimensions (i.e., "leadership", "people", "strategy", "partnerships and resources", and "processes, products and services") and the "results" area with four dimensions (i.e., "people results", "customer results", "society results", and "business results"). "Enablers" describe what an organization should do and how to achieve its organizational goals. "Results" focus on what is important to the key stakeholders. From the perspective of the evaluation dimension, the EFQM excellence model contains organizational enablers (how) and the results of project performance (what), which can be used to evaluate project success and to measure and improve the performance of a project [22].

Previous studies presented that the EFQM excellence model framework and ideas were used in the construction industry. Vukomanovic et al. [47] developed an evaluation method by combining the EFQM excellence model and the balanced scorecard with setting weights by the analytic hierarchy process (AHP) to realize strategic control for construction enterprises. This method was validated by analyzing data from 32 construction companies in southeastern Europe, and the result also showed that the EFQM excellence model has high applicability to contractual businesses. Mohammad Zadeh [48] used the EFQM excellence model to measure the excellence degree of a construction firm, and the research results indicated that the external environment of the construction firms would change at different stages of the project. Chileshe [49] analyzed the impact and correlation of the "enabler" and the "result" criteria via structural equation modeling on the data from construction firms, and then proposed a modified model fitting for construction firms. Oladinrin and Ho [50] used the EFQM excellence model to provide strategic support to the construction ethics of companies. Shanmugapriya and Subramanian [51] calculated the quality performance of construction projects by using structural equations based on the EFQM excellence model, which reflected the interrelationship among the various criteria.

Additionally, the industrialization and lean construction of projects are highly similar in terms of their management objectives to eliminate waste and uncertainty in the building construction process [52]. Nesensohn et al. [53] proposed a lean construction maturity evaluation framework based on the EFQM excellence model via brainstorming, expert interviews, and questionnaires. Oakland and Marosszeky [54] likewise developed a new lean quality model that provides a simple framework for excellent performance, thereby covering all aspects of the project organization and its operation. It has been proven that the EFQM excellence model, as a "business excellence" approach, played a significant role in improving the performance of the construction projects. It has also been proven that employing the EFQM excellence model for the project maturity evaluation was applicable, especially supporting performance improvement by considering the organizational enablers.

### 2.4. Summary

The motivation of this study was to develop a new evaluation system to reveal where the weakness lies as well as the roadmap to implement the performance improvements. To explore the solution to establish the new evaluation method, we confirmed the CMM theory and the EFQM excellence model as the theoretical basis of this study, because the above-related work indicated that: (1) The CMM is widely used to evaluate project performance and can provide ideas to clarify the present situation and

the development potential of the construction industrialization, and (2) the EFQM excellence model considers roundly including what needs to be improved and how to implement improvements by the organization. Considering the existing gap between project-based IC maturity evaluation and the perspective of project governance, this study aimed to establish an ICMM by integrating the CMM theory and the evaluation ideas of the EFQM excellence model to reveal the status of IC projects and confirm the performance improvement path from the perspective of facilitating the organization. Specifically, the EFQM guides the formation of the basic framework of the ICMM, and the evaluation system of the presented study is determined according to the construction process of the CMM.

## 3. Research Methods

The research methods in this study involved the establishment and validation of the ICMM.

The ICMM establishment was implemented as the following three steps: Setting the evaluation indicator system, weighting the indicators, and designing maturity levels. First, the evaluation indicator system: The evaluation indicators were determined by a literature review and face-to-face semi-structured interviews. The participants of the interviews were 15 experts with abundant IC cognition and practical experiences from a real estate development agency, construction contracting firm, architectural design company, prefabricate component factory, and research institutes, respectively. Second, indicator weight determination: The weights of the confirmed indicators were assigned by the expert scoring method and the AHP [55]. Third, maturity level design: The maturity levels were determined by employing the typical maturity model, and standard text research combined with the practical scenarios of the IC mode implementation.

A multi-case study was conducted to validate the ICMM. Four typical under-construction industrialized building projects located in Shanghai were selected. A questionnaire survey with evaluation items to rate the projects and the face-to-face semi-structured interviews for the model evaluation were carried out. This method has been previously used for the evaluation of a maturity model for supply chain relationships in construction [39]. All interviewees were provided with information on the ICMM and the evaluation procedures before the interviews. Five questions were asked in each interview, respectively, involving the appropriateness of the evaluation dimensions, indicators, practical evaluation items, maturity levels, and whole evaluation procedure. Ultimately, the applicability and reliability of the ICMM was evaluated by analyzing the scoring results of the selected projects and the direct results of the face-to-face semi-structured interview. This study invited 10 managers (construction practitioners) from these 4 selected projects to participate in the model evaluation interview and two senior engineers in each project were selected to participate in rating the corresponding project that they were involved in. The specific information of the survey participants is presented in Section 5.

## 4. ICMM Establishment

According to the CMM theory, the rating components of the ICMM, to assess the IC maturity of a building project, are its maturity levels, key evaluation areas, and their goals. This section specifically describes the following three steps to establish ICMM: (1) Evaluation indicator system determination, (2) indicator weight determination, and (3) maturity level design.

### 4.1. Evaluation Indicator System Determination

The evaluation indicator system of the ICMM mainly employed the framework of the EFQM excellence model via a top-down approach. This system should retain the superiority of the EFQM excellence model and sufficiently adapt to the scenario of the IC projects. The EFQM excellence model is primarily used for enterprises with continuous businesses, as several differences of the organizational structure, operation activities, and operation goals exist in a construction project due to the temporary organization consisting of multiple participants. Therefore, the indicator system of the ICMM was also divided into two key areas by referring to the EFQM excellence model, namely, "enablers" and

"results"; adjustments on evaluation dimensions were made to adapt to the evaluation of the IC projects; and the indicators were subsequently identified by inducing, classifying, removing, and keeping the critical indicators from the literature review or practical interviews.

First, evaluation dimension identification: The "enablers" area consists of four main evaluation dimensions associated with the organizational enablers (i.e., process facilitator and discursive ability [56]), namely, "leadership" "participant capabilities, and collaboration", "planning and control", and "technology and schema". Given the temporality of project organizations, the "strategy" factor existing in the EFQM excellence model mainly indicating long-term development planning was not considered in this study. The "results" area consists of four main indexes involved in outcome factors (i.e., building entity, practical implications, organizational growth, and project process), namely, "product", "society", "organization", and "management and control". From the "results" area, the organization can learn from the outcome performance of the projects and obtain innovative ideas. Figure 1 shows the initial framework of ICMM to the main index level, and the adjustment details of the main indexes are shown in Tables 3 and 4.

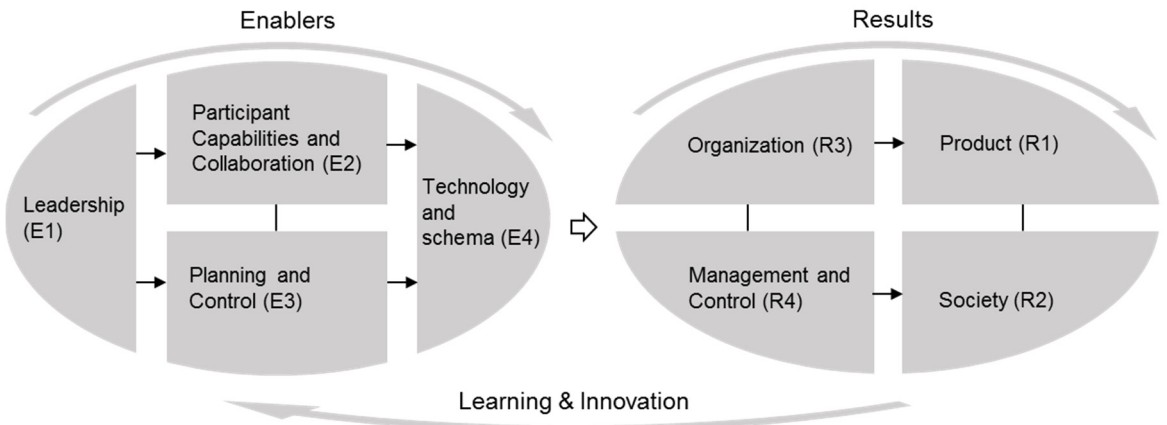

**Figure 1.** An initial framework for the industrialized construction maturity model.

Second, indicator identification: According to the identified evaluation dimensions, the indicators were identified by reviewing the related literature, existing documents, and interview results. Specifically, indicators in the "enablers" area that involve organizational enablers were identified by inducing the critical success factors, barriers, and driver factors of IC [7,57–60], and were replenished by summarizing the results of the expert interviews from practical projects. Meanwhile, indicators in the "results" area that involve the outcome performance were identified by summarizing the existing IC evaluation standards in China, and by adding the missing indicators from the EFQM excellence model and the relevant literature. Tables 5 and 6 present the indicators in the "enablers" area and the "results" area, respectively.

**Table 3.** Evaluation dimensions in the "enablers" area of the ICMM.

| Code | Dimensions | Organizational Enablers | Justification for Adjustment from the EFQM Excellence Model |
|---|---|---|---|
| E1 | Leadership | Discursive ability (1): Project participants have the basic cognition of the IC mode and are preparing to take practical action to promote performance realization and optimization of IC projects. | All the participants have been integrated into one temporary team, the leaders refer to the project owner and leaders of other participants, who reach the agreements based on a series of contracts. "Leadership" encompasses the cognition and attitudes of the participants to the IC. |
| E2 | Participants' capabilities and collaboration | Discursive ability (2): The project organization consists of multiple participants, of which capabilities and experience need to meet requirements of project implementation and promote participants to work together efficiently. | Given that project organizational members are not individuals but project participants, the dimension "People" in EFQM is introduced to the ICMM, namely, "Participants' capabilities and collaboration". |
| E3 | Planning and control | Process facilitator (1): Project participants should take preparation to promote performance realization and optimization of IC projects. | The dimension of "processes, products, and services" in EFQM is adjusted into "planning and control" to represent the organizational process of IC projects, which involves the project goal setting and process control. |
| E4 | Technology and schema | Process facilitator (2): The project needs to select effective technologies and rational schemas to deal with the industrial obstacles and difficulties of the IC projects in design, production and construction stages. | The organization needs to make decisions for selecting appropriate techniques and schemas to implement the IC mode, and these decisions involving technical management, information, and knowledge management can be comprehensively summarized as the dimension in terms of "technology and schema". |

**Table 4.** Evaluation dimensions in the "Results" area of the ICMM.

| Code | Dimensions | Outcome Factors | Justification for Adjustment from the EFQM Excellence Model |
|------|------------|-----------------|-----------------------------------------------------------|
| R1 | Product | Project entity: The final building entity for delivery with its associated performance, status. | The owner of an IC project is considered as a customer, and the constructed building entity is the product; given only the construction stages considered in ICMM, "Customer" in EFQM is adjusted to the "product". |
| R2 | Society | Social performance: Practical benefits to society resulting from project delivery of final products, such as energy conservation and environmental protection, technical innovation, etc. | Given the same meaning of the original evaluation dimension of "Society" in EFQM with that in IC projects, it is unchanged and indicates the practical implications to society. |
| R3 | Organization | Organizational performance: The satisfaction of project participants with communication, cooperation and growth of the organization itself. | The project organization contains both individuals and participants from different companies, after going through a complete project, the organization should have a certain amount of growth. |
| R4 | Management and control | Project process performance: The degree of success of the project to achieve the three objectives (quality, schedule, cost) of construction project management. | Since the ultimate goal of a long-term organization is an achievement on business and the main project objective is to reach the one-off project delivery. The "business" index is adjusted into "management and control". |

**Table 5.** Indicators in the "enablers" area of the ICMM.

| Dimensions | Code | Indicators | Definitions | Justification |
|---|---|---|---|---|
| Leadership (E1) | E1.1 | Owner's cognition and attitude | The degree of the project owner to recognize and support the IC mode. | Client skepticism and resistance [59]. Clients suspicious about performance but build a good location for higher prices [58]. |
| | E1.2 | Contractors' cognition and attitude | The degree of project contractors to recognize and support the IC mode. | Lack of confidence in offsite production in the industry [59]. Conservative industry culture [57]. Reluctance of manufacturers to innovate and change to MMCs; Mindset of the industry (cultural problems) [58]. |
| Participants' capabilities and collaboration (E2) | E2.1 | Designer's experience and ability | The experience and ability of the project designer to implement the design of IC projects and deal with the technical problems of the IC mode. | Interview results: The design unit has experience in construction industrialization, and those directly involved in the design have sufficient industrial design experience to ensure the design quality. |
| | E2.2 | Construction contractor's experience and ability | The experience and ability of the project contractor to implement the construction work of IC projects and deal with the on-site process and management problems of the IC mode. | Contractor leadership; Contractor experience [7]. Lack of previous experience and guidance; Higher skill demands for the labor [59]. Lack of experience and skills [58]. Availability of qualified structural engineers specialized in precast concrete systems; Availability of contractors specialized in precast concrete systems; Availability of laborers specialized in precast concrete systems [60]. |
| | E2.3 | Component supplier's experience and ability | The experience and ability of the component supplier to implement the component production of IC projects and deal with the transportation and site-assembly problems. | Module Fabricator Capability [7]. Manufacturing capacity [59]. Limited capacity of existing manufacturers [58]. |
| | E2.4 | Cooperation willingness | The willingness of participants to collaborate by taking the initiative to share information and communicate with each other. | Alignment on Drivers; Vendor Involvement [7]. Design-bid-build contracts split design and production [57]. Owners' capability of providing good communication among parties [60]. |
| | E2.5 | Collaboration channels | Effectiveness and diversity of cooperation among participants | Poor integration for the supply chain [59]. Inadequate coordination: procurement, supply chain, site management; Low IT integration in the industry [58]. |

**Table 5.** *Cont.*

| Dimensions | Code | Indicators | Definitions | Justification |
|---|---|---|---|---|
| Planning and control (E3) | E3.1 | Goal setting | Whether clear and reasonable project goals (schedule, quality, cost, etc.) have been set in advance. | Interview results: In the project process, clear and reasonable prefabricated construction goals are gradually established (application standards, schedule, quality, cost plan). |
| | E3.2 | Norms and standards | The current application of IC specifications and standards on the project, and whether project-specific process standards have been developed. | Lack of available codes and standards [59]. STA norms and rules; Governmental rules regarding plans [57]. Fewer codes/standards available; Regulatory authorities: not yet included in planning regulations [58]. |
| | E3.3 | Schedule control | The organization's control effect on the project progress measured by checking key nodes and taking measures to avoid delays in the construction period. | Early Completion Recognition; Owner Delay Avoidance; Continuity through Project Phases; Transport Delay Avoidance [7]. Longer lead-in time [59]. |
| | E3.4 | Change control | The organization's control effect on the planning implementation measured by checking the difference between the design schema and construction results. | Timely Design Freeze [7]. The inability to freeze the design early on [59]. Early design freeze, due to the long lead-in time, and extensive planning; Inflexible/not suitable for late design changes [58]. |
| | E3.5 | Quality control | The organization's control effect on quality monitoring measured by checking the difference between construction results between proven quality criteria in the construction industry. | Less tolerance between factories made components and on-site assembly; Lack of quality assessment tools and accreditation [58]. |
| | E3.6 | Cost control | The organization's control effect on the project cost by identifying the factors that reduce the cost to insure the total cost within the budget. | Cost Savings Recognition [7]. |
| Technology and schema (E4) | E4.1 | Prefabricated technology system | The overall technical solution to achieve the purpose of IC includes structural modularization, prefabrication ratio, component design, installation process design, etc. | Owner- Furnished/Long Lead Equipment Specification [7]. Highly restrictive construction tolerances [59]. (Component) lack of large-scale and repetition possibilities [57]. Poor integration and interface performance with traditional method [58]. Variety of precast concrete components; Conformity between different precast concrete systems [60]. |

**Table 5.** *Cont.*

| Dimensions | Code | Indicators | Definitions | Justification |
|---|---|---|---|---|
| Technology and schema (E4) | E4.2 | Advance work of IC design | The organization's effect on considering more on-site assembly and construction in the design phase | Interview results: Early design can consider the requirements of component splitting in later design to meet the requirements of industrial building system in production, transportation and construction |
| | E4.3 | Design with component confirmation | The organization's effect on structural splitting and determination of prefabricated components | Interview results: Structural design can use regular and batch components as much as possible to reduce the appearance of special-shaped components |
| | E4.4 | Detail design and process matching | The organization's effect on deepening the process design of participants' cooperation, and deepening the design and construction to meet the requirements of the component production. | Interview results: Relevant parties in detailed design can provide design auxiliary information in time, and detailed design delivery documents meet the production needs of component factories |
| | E4.5 | Component production | The efficiency and quality of producing components in the factories. | Interview results: Component suppliers use a more efficient and environmentally friendly manufacturing process, and component capacity needs to meet project needs |
| | E4.6 | Component transportation | The efficiency and safety of component transportation by considering the distance, component protection, etc. | Module Envelope Limitations; Transport Infrastructure [7]. Transportation [59]. Expensive long-distance transportation for large and heavy loads [58]. Size and load restrictions on transportation [60]. |
| | E4.7 | Construction with component assembly | The efficiency, quality and safety of component assembly and integral construction on site | Heavy Lift/Site Transport Capabilities [7].Specific demands for the site logistics for pre-finished elements protection [59]. Site-specific constraints, e.g., access limitations and space for large loads [58]. |
| | E4.8 | Industrialized decoration | The organization's effect on considering the modular decorating components to meet users' demands. | Interview results |
| | E4.9 | Operation preparation | The organization's effect on preparing for operation and maintenance in the early stage. | Operations and Maintenance (O&M) Provisions [7]. Increase in complexity for maintenance [59]. |

**Table 6.** Indicators in the "results" area of the ICMM.

| Dimensions | Code | Sub-Index | Description | Justification |
|---|---|---|---|---|
| Product (E1) | R1.1 | Prefabrication rate | An important indicator for measuring the degree of assembly of a unit's building structure. | SAPB_GBT51129-2017 [61] |
| | R1.2 | Practical performance | The degree to which the completed building meets actual functional needs. | SAPB_GBT51129-2017 [61]; interview results: users' satisfaction with the project depends on its conformity with the technical standards. |
| | R1.3 | Operating and maintenance savings | The degree of savings achieved by the IC model for the operation and maintenance phase of the completed building. | Pan et al. [62]; interview results: Due to reasonable product design and quality clearance, users save maintenance costs compared to the market average product during long-term use. |
| | R1.4 | Owner's satisfaction | Owner's overall satisfaction with the completed project. | EFQM [63] |
| Society (R2) | R2.1 | Technological innovation | The degree and effect of the project's efforts in technological innovation during the implementation process. | EFQM [63] |
| | R2.2 | Environmentally friendly | The effect of energy-saving, water-saving, material saving, and environmental protection measures in the process of Project Construction. | SAPB_GBT51129-2017 [61] |
| | R2.3 | Honors or awards | The status of the project's social recognition, such as industrial housing pilot project, demonstration project. | EFQM [63] |
| Organization (R3) | R3.1 | Participants communication efficiency | Project participants' effect on information exchange and communication with each other. | EFQM [63] |
| | R3.2 | Participants' long-term cooperation willingness | Project participants' satisfaction with project cooperation, profit, and willingness to cooperate with other participants again. | EFQM [63]; Interview results: experts suggest that the long-term willingness of the participants to cooperate can better reflect whether the participants are satisfied with the project's cooperation method and risk and benefit allocation. |
| Management and control (R4) | R4.1 | Schedule | Used to evaluate the results of schedule control: whether there is a delay compared to the originally planned duration. | ESIH_DG/TJ 08-2198-2016 [64] |
| | R4.2 | Quality | Used to evaluate the results of quality control: whether the project quality meets the requirements of the corresponding specifications. | ESIH_DG/TJ 08-2198-2016 [64] |
| | R4.3 | Cost | Used to evaluate the results of cost control: whether the project cost is controlled within the budget. | ESIH_DG/TJ 08-2198-2016 [64] |

Third, practical evaluation item identification: The evaluation items corresponding to the evaluation indicators directly evaluate the state of an IC project. In all, 57 evaluation items of the "enablers" area and 15 evaluation elements of the "results" area were used. During the rating process, all items of the scale in a questionnaire were measured using the Likert 5-point scale. This scaling approach is concise and easy to answer and is widely used in tools to measure respondents' views, beliefs, and attitudes. The degree of conformity between the description and the actual situation in the rating questionnaire was carried out by the five levels of "extremely conformity", "reluctantly conformity", "uncertain", "unconformity", and "extremely non-conformity". The questionnaire can be found in Supplementary Materials section.

### 4.2. Indicator Weight Determination

Indicators in the whole evaluation system have various weights. The AHP approach [55] was used to calculate the weights of the indicators according to the characteristics of sample data and recommendations from previous studies [65,66]. Figure 2 presents the AHP process, as specifically shown in the following five steps.

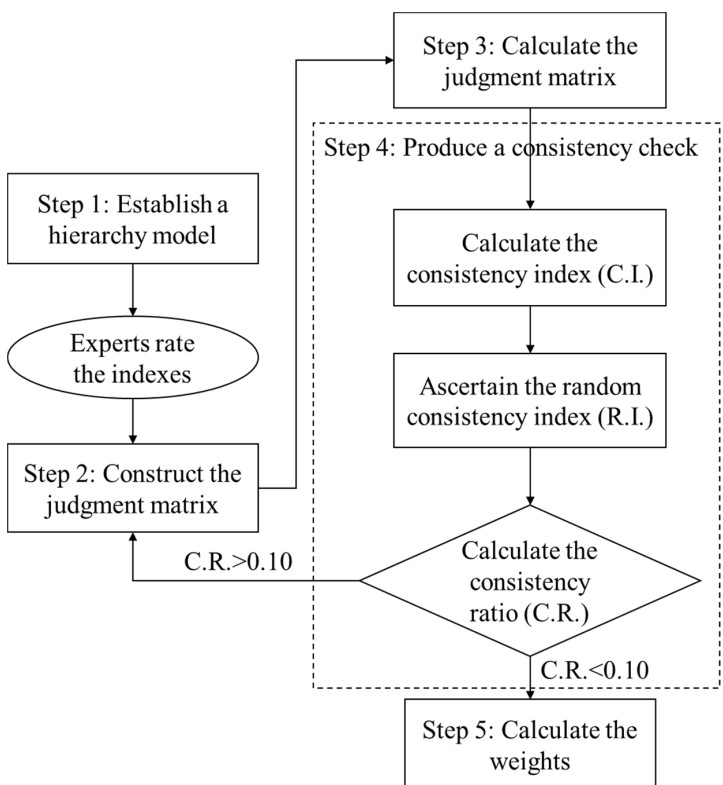

**Figure 2.** Steps of the analytic hierarchy process.

Step 1: The hierarchical structure of the system was established by the AHP approach, which was confirmed based on the indicator system in Section 4.1. The target layer corresponds to the "enablers" and "results" areas, respectively. The criterion layer corresponds to the evaluation dimensions, and the index layer corresponds to the indicators.

Step 2: The judgment matrix A was constructed (two judgment matrices in this research indicating the "enablers" and "results" areas). This step entailed utilizing a scale ranging from 1 to 9 to measure the results of pairwise comparisons and setting priorities on every level of the hierarchy according to the organized structure given by Step 1 [65]. In this step, the coefficient importance scores of all indexes were determined by expert scoring. As shown in Table 7, 15 experts, including project owners,

contractors, consultants, and scholars, answered the index weighting questionnaire. The final rating scores are presented in Tables 8 and 9. Then, the judgment matrix A is constructed.

**Table 7.** Experts' information for weighting questionnaire.

| No. | Institution Type | Company Type | Position |
|---|---|---|---|
| 1 | Real estate developer | Private | Project manager |
| 2 | Designer | Public | Department director |
| 3 | Designer | Public | Department deputy director |
| 4 | Designer | Public | Department deputy director |
| 5 | Designer | Public | Department staff |
| 6 | Designer | Public | Department staff |
| 7 | Component factory | Private | Head of the team |
| 8 | Component factory | Public | Head of the team |
| 9 | Construction unit | Public | Head of the assembly building research center |
| 10 | Construction unit | Public | Head of the team |
| 11 | Construction unit | Private | Head of the team |
| 12 | Assembly consulting | Private | Corporate executive director, design director |
| 13 | Site supervisor consulting | Private | Technical specialist |
| 14 | University | Public | Scholar (Ph.D.) |
| 15 | University | Public | Scholar (Ph.D.) |

**Table 8.** Consistency test results of indicators in the "enablers" area.

| Dimensions | Mean Sig. | Code | Indicators | Mean Sig. | C.I. | C.R. |
|---|---|---|---|---|---|---|
| E | | | | | 0.000 | 0.000 |
| Leadership (E1) | 4.1538 | E1.1 | Owner's cognition and attitude | 3.9231 | 0.000 | 0.000 |
| | | E1.2 | Contractors' cognition and attitude | 3.5385 | | |
| Participants' capabilities and collaboration (E2) | 4.3077 | E2.1 | Designer's experience and ability | 4.3846 | 0.014 | 0.012 |
| | | E2.2 | Construction contractor's experience and ability | 4.0769 | | |
| | | E2.3 | Component supplier's experience and ability | 4.0769 | | |
| | | E2.4 | Cooperation willingness | 4.0000 | | |
| | | E2.5 | Collaboration channels | 3.9231 | | |
| Planning and control (E3) | 4.1538 | E3.1 | Goal setting | 4.0000 | 0.016 | 0.013 |
| | | E3.2 | Norms and standards | 4.0769 | | |
| | | E3.3 | Schedule control | 4.0000 | | |
| | | E3.4 | Change control | 3.9231 | | |
| | | E3.5 | Quality control | 3.8462 | | |
| | | E3.6 | Cost control | 4.0000 | | |
| Technology and schema (E4) | 4.3077 | E4.1 | Prefabricated technology system | 4.2308 | 0.004 | 0.004 |
| | | E4.2 | Advance work of IC design | 4.1538 | | |
| | | E4.3 | Design with component confirmation | 3.9231 | | |
| | | E4.4 | Detail design and process matching | 3.7692 | | |
| | | E4.5 | Component production | 3.6923 | | |
| | | E4.6 | Component transportation | 3.6923 | | |
| | | E4.7 | Construction with component assembly | 3.9231 | | |
| | | E4.8 | Industrialized decoration | 3.6923 | | |
| | | E4.9 | Operation preparation | 3.6923 | | |

**Table 9.** Consistency test results of indicators in the "results" area.

| Dimensions | Mean Sig. | Code | Indicators | Mean Sig. | C.I. | C.R. |
|---|---|---|---|---|---|---|
| R | | | | | 0.015 | 0.017 |
| Product (R1) | 4.2308 | R1.1 | Prefabrication rate | 3.6154 | 0.007 | 0.008 |
| | | R1.2 | Practical performance | 4.3077 | | |
| | | R1.3 | Operating and maintenance savings | 3.9231 | | |
| | | R1.4 | Owner's satisfaction | 4.2308 | | |
| Society (R2) | 3.9231 | R2.1 | Technological innovation | 3.9231 | 0.027 | 0.046 |
| | | R2.2 | Environmentally friendly | 4.0000 | | |
| | | R2.3 | Honors or awards | 3.8462 | | |
| Organization (R3) | 3.3077 | R3.1 | Participants communication efficiency | 4.3077 | 0.000 | 0.000 |
| | | R3.2 | Participants' long-term cooperation willingness | 4.2308 | | |
| Management and Control (R4) | 3.8462 | R4.1 | Schedule | 4.0769 | 0.000 | 0.000 |
| | | R4.2 | Quality | 4.3846 | | |
| | | R4.3 | Cost | 4.0769 | | |

Step 3: The judgment matrix A is calculated. The eigenvalues and eigenvectors that satisfy Equation (1) were calculated:

$$AW = \lambda_{max}W, \tag{1}$$

where $\lambda_{max}$ is the largest eigenvalue and $W$ is the normalized eigenvector corresponding to $\lambda_{max}$. The component $i$ of vector W, which is denoted as $w_i$, is the weight of the corresponding element according to the sorting result. The objective of this step was to find the maximum eigenvalue "$\lambda_{max}$" [55].

Step 4: A consistency check was produced, which represents a critical step. A consistency check for the decision matrix was performed as follows:

First, the consistency index (*C.I.*) was calculated as Equation (2):

$$C.i. = \frac{\lambda_{max-n}}{n-1}, \tag{2}$$

where $n$ is the matrix size. The smaller the *C.I.* value, the smaller the deviation from the consistency. The consistency in the judgments of the relative importance of attributes reflects the cognition of the analyst.

Second, the random consistency index (*R.I.*) [55] was ascertained according to Table 10.

**Table 10.** Values of *R.I.*

| N | 1 | 2 | 3 | 4 | 5 | 6 | 7 | 8 | 9 | 10 |
|---|---|---|---|---|---|---|---|---|---|---|
| R.I. | 0 | 0 | 0.58 | 0.90 | 1.12 | 1.24 | 1.32 | 1.41 | 1.46 | 1.49 |

Third, the consistency ratio (C.R.) was calculated as Equation (3):

$$C.R. = \frac{C.I.}{R.I.}. \tag{3}$$

The consistency of the judgment matrix is generally considered as acceptable when *C.R.* < 0.10. The results of the consistency test are also presented in Tables 8 and 9.

Step 5: The weights were calculated by completing Steps 1 to 4 for all levels in the hierarchy (i.e., dimensions and indicators) of the ICMM. The weight of each indicator was determined based on the comparison with indicators in the same dimension after determining the weight of each dimension separately. Specifically, in the "enablers" area, the weights of evaluation dimensions $P_i$ ($i$ = $E1$, $E2$, $E3$, and $E4$) and the relative weights of indicators to their corresponding evaluation dimension $R_{ij}$ (e.g., $R_{12}$ means the relative weight of E1.2 to E1) were first determined separately. Then, the final weight of the indicator $P_{ij}$ was the weight of its corresponding dimension $P_i$ multiplied by the relative weight of the indicator $R_{ij}$. The weighting calculation rules and results (%) of the indicators are presented in Figure 3.

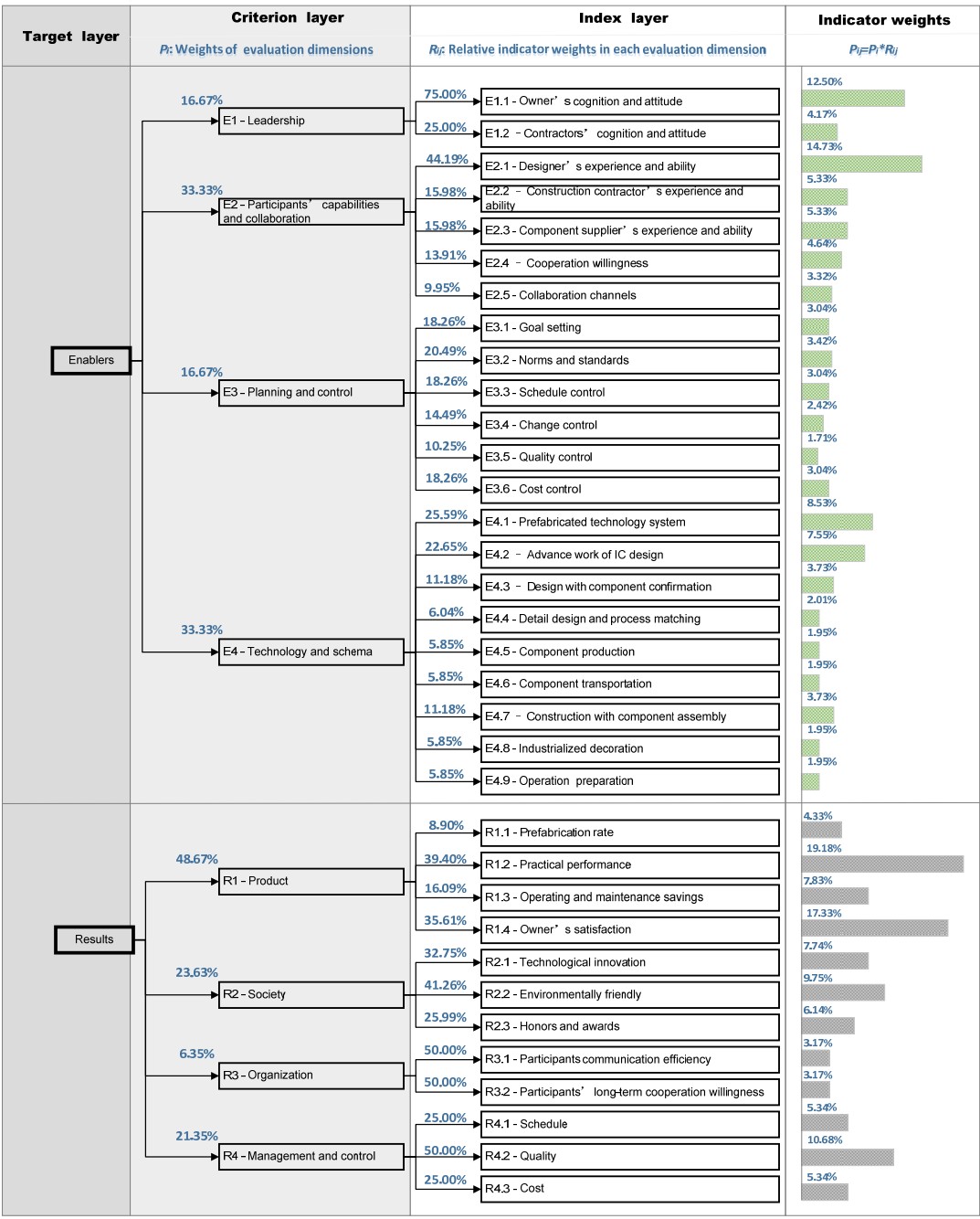

**Figure 3.** Weights of ICMM indicators.

### 4.3. Maturity Level Design

Similar capability maturity models have three to five levels [15], and four maturity levels have been proven to be sufficient to reflect the full spectrum of the IC maturity growth process [28,29]. This study draws on the building information modeling (BIM) maturity evaluation [67] and proposes the draft of the IC project maturity rating, which is divided into four levels (see Figure 4), namely, the initial level, the upgraded level, the integrated level, and the optimal level. Since the final scores are all values of 1–5, and the full score is 5, the score divided by 5 is the percentage corresponding to the final maturity grade. Tables 11 and 12 give the specific explanations of the maturity levels of each evaluation dimension in the "enablers" and "results" areas, respectively. The general status of IC projects with different maturity levels is described below.

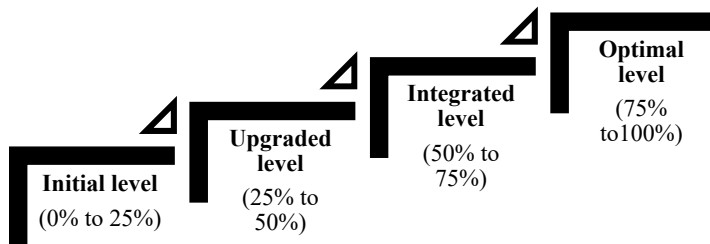

**Figure 4.** Industrialized construction maturity levels.

The initial level (0% to 25% included the project participants initially applying the IC technologies, the relevant process, and deploying the public standard of IC. The participants' understanding of IC is insufficient, and the industrialization is not professional enough. Moreover, the project benefits created by industrialization are small.

The upgraded level (25% to 50%) implies that the project participants' recognition towards the IC mode has been significantly raised. The project organization mainly focuses on selecting appropriate technologies and schemas and can make considerable efforts to improve the performance involving the environment, IC technology application, and the organization itself. The technical experience of IC begins to accumulate.

Integrated level (50% to 75%): Here, the project under the more mature IC mode has a clear criterion for IC technology selection, process schema selection, and a performance evaluation basis. Participants start to use information technology (e.g., BIM, information management system) for collaborative cooperation, technical performance, cross-organizational performance adjustment, and optimization. The IC value is gradually reflected in the final performance of the project.

Optimal level (75% to 100%): The project participants have a very clear recognition of IC and take initiative in exploring more mature technical solutions for performance improvement. Information management technologies (e.g., BIM) are applied manually to achieve the integration of the IC supply chain, standardization, and automated operations. Management programs continue to optimize the mechanism.

**Table 11.** Description of maturity levels of each evaluation dimension in the "enablers" area.

| Dimensions | Initial Level | Upgraded Level | Integrated Level | Optimal Level |
|---|---|---|---|---|
| Leadership (E1) | Project participants have little cognition of the IC mode and have few willingness to learn to adopt IC technologies. | Project participants have a basic cognition of the IC mode and are willing to explore the IC applications. | Project participants have the basic cognition of the IC mode and are strongly willing to implement the IC applications. | Project participants have sufficient cognition of the IC mode and have a very strong willingness to achieve the IC applications. |
| Participants" capabilities and collaboration (E2) | The organization structure hardly meets the IC projects' requirements and the participants can work together reluctantly. | The organization structure reluctantly meets the IC projects' requirements and the participants can work together preliminarily. | The organization structure mostly meets the IC projects' requirements and the participants can work together efficiently. | The organization structure fully meets the IC projects' requirements and the participants can work together very closely. |
| Planning and control (E3) | Project managers make rough planning for key progress prediction and resource allocation and take a few measurements to control the implementation process of the planning. | Project managers make rough planning for key progress prediction and resource allocation and take some rational measurements by manual to control the implementation process of the planning. | Project managers make detailed planning for key progress prediction and resource allocation and take some rational measurements by manual to control the implementation process of the planning. | Project managers make very detailed planning for key progress prediction and resource allocation and take some information tools (BIM, Integrated information platform, etc.) to control the implementation process of the planning. |
| Technology and schema (E4) | The organization has few experiences in selecting technologies and process schemas to deal with the industrial obstacles and difficulties of the IC mode in components design, production, transportation, and assembly. | The organization has the basic experiences in selecting technologies and process schemas to deal with the industrial obstacles and difficulties of the IC mode in components design, production, transportation, and assembly. | The organization has rich experiences in selecting technologies and process schemas to deal with the industrial obstacles and difficulties of the IC components design, production, transportation, and assembly. | The organization has rich experiences in both implementing risk management measurements and selecting technologies and process schemas to deal with the industrial obstacles and difficulties of the IC components design, production, transportation, and assembly. |

**Table 12.** Description of maturity levels of each evaluation dimension in the "results" area.

| Dimensions | Initial Level | Upgraded Level | Integrated Level | Optimal Level |
|---|---|---|---|---|
| Product (R1) | The building entity for the final delivery barely meets the owner's requirements, and the appearance and function of the building reach the design goal and meet the basic needs of practical applications. | The building entity for the final delivery mostly meets the owner's requirements, and the appearance and function of the building reach the design goal and meet key needs of practical applications reluctantly. | The building entity for the final delivery fully meets the owner's requirements, and the appearance and function of the building achieved the design goal and meet needs of practical applications exactly. | The building entity for the final delivery fully meets the owner's requirements, and the appearance and function of the building achieved the design goal very well and meet all needs of practical applications sufficiently. |
| Society (R2) | The benefits of the construction process brought by the IC mode have no difference from the traditional construction mode. | The construction process under the IC mode is more environment-friendly than the traditional construction mode. | The IC mode brings more energy conservation and makes the construction process more environment-friendly than the traditional construction mode. | The IC mode brings more technology innovation and more energy conservation, as well as more environment friendly than the traditional construction mode. |
| Organization (R3) | The inner cooperation of participants can make it possible to reach the IC project objectives reluctantly, but the efficiency of the cooperation process is low. | The inner cooperation of participants can support achieving the project objectives well, and the efficiency of the cooperation process is very smooth. | The inner cooperation of participants can support achieving the project objectives well, the cooperation process is very smooth, and the member within the organization can learn to grow. | The inner cooperation of participants can support achieving the project objectives well, the cooperation process is very smooth, the member within the organization can learn to grow, and the participants have the willingness to establish a long-term cooperative relationship. |
| Management and control (R4) | The management and control of the construction process under the IC mode can barely achieve the three major objectives (quality, schedule, or cost). | The management and control of the construction process under the IC mode make the construction process are equal to the traditional mode in terms of quality, progress, or cost. | The management and control of the construction process under the IC mode make the construction process slightly better in terms of quality, progress, and cost. | The management and control of the construction process under the IC mode make the construction process much better in terms of quality, progress, and cost. |

## 5. ICMM Validation

The ICMM validation was conducted by a multi-case study with real-life IC project maturity evaluation and face-to-face semi-structured interviews for model evaluation. First, four typical IC building projects under construction from Shanghai were selected. Second, the IC maturity level of the selected projects was determined by questionnaire scoring. Third, the face-to-face semi-structured interviews were conducted, where the projects' maturity evaluation results were discussed and the ICMM was evaluated. The applicability and operability of the ICMM can be demonstrated according to the multi-case study and the expert interviews.

### 5.1. Background of the Multi-Case Study

Table 13 gives the basic information of the selected typical prefabricated construction projects under construction from Shanghai, which is one of the first cities in China to fully enforce prefabricated buildings. The four selected projects differ in the delivery method, investment type, construction structure type, and organizational management process. For example, in terms of the project delivery method, projects A and D adopt design-building (DB) and design-bid-building (DBB), respectively, while projects B and C adopt the method of construction management at a general contractor (CM-at GC) [68]. In terms of the building type, projects A and B are multi-story public buildings, and projects C and D are high-rise residential projects. In terms of investment type, projects A and D are both public investment projects, and projects B and C are both private investment projects. In terms of the organizational management process, there are significant differences in whether these projects have developed specific IC implementation standards and whether to use information technology and tools (e.g., BIM, or information management software platform) as support. All these specific differences have different impacts on the final maturity evaluation results, as detailed in the results of the subsequent evaluation results discussion part. Given the diversity of the above, the selected projects represent typical IC construction projects in China to validate the ICMM.

**Table 13.** Basic information about four selected typical IC projects.

| 0 | Project Code | A | B | C | D |
|---|---|---|---|---|---|
| 1 | Project Name | *** Middle School | ***Plaza | *** Housing Project | *** resettlement houses |
| 2 | Delivery method | DB | CM-at GC | CM-at GC | DBB |
| 3 | Investment type | Public | Private | Private | Public |
| 4 | Construction scale (m$^2$) | | | | |
| 4.1 | Aboveground area | 39,269.07 | 99,793.09 | 69,954.73 | 102,987.00 |
| 4.2 | Underground area | 9924.00 | 49,876.78 | 28,109.97 | 432,020.00 |
| 5 | Structure Type | | | | |
| 5.1 | Building Type | Multi-story public building | Multi-story public building | High-rises residence | High-rises residence |
| 5.2 | IC Technology | Precast concrete framework | Precast concrete framework | PC-integral shear wall structure | PC-Composite Shear Wall |
| 6 | Distance from component factory | 150 km | 60 km | 50 km | 3 km |
| 7 | Process standard | | √ | | |
| 8 | Informatics techniques | | | | |
| 8.1 | BIM application | √ | √ | √ | |
| 8.2 | BIM application stage | Construction | Design and construction | Design | – |
| 8.3 | Information platform | √ | | | |

To ensure the scoring process was reliable, we selected two practical participants from each project who had a full understanding of the IC practice at the evaluation item level. The eight raters, together with the other three project practitioners, also participated in the interview, and then the ICMM itself and its application process were analyzed and evaluated. The 11 respondents were all from different participants and had more than 5 years of experience in IC projects and could objectively evaluate the industrialization maturity of the project. Table 14 gives the basic information of the survey respondents (including interviewees and project raters).

**Table 14.** Basic information of respondents (including interviewees and project raters).

| Code | Gender | Age | Position | Participants | Experience * | | | Rate | Interview Time (min) |
|------|--------|-----|----------|--------------|------|-------|------------|------|------------|
| | | | | | Year | Count | Scale (km$^2$) | | |
| A1 | Male | 35–45 | Scholar | Construction contractor | 10 + | 1 | 49 | √ | 68 |
| A2 | Male | 25–35 | Manager | Construction contractor | 10+ | 3 | 100~200 | √ | 69 |
| B1 | Male | 35–45 | Manager | Construction contractor | 5–10 | 3 | 100~150 | √ | 61 |
| B2 | Male | 35–45 | Manager | Design | 10+ | 3 | 100~150 | √ | 51 |
| C1 | Male | 25–35 | Manager | Owner | 5–10 | 2 | 50~100 | √ | 90 |
| C2 | Female | 25–35 | Manager | Design | 10+ | 9 | 60~200 | √ | 72 |
| C3 | Male | 35–45 | Manager | Construction contractor | 10+ | 1 | 100 | | 35 |
| C4 | Male | 25–35 | Manager | Labor contractor | 5–10 | 5 | 50~100 | | 46 |
| D1 | Male | 35–45 | Manager | Owner | 10+ | 1 | 145 | √ | 92 |
| D2 | Male | 35–45 | Manager | Design | 10+ | 12 | 30~200 | √ | 89 |

*. The experience of the respondents includes three aspects: the time of employment (year), the number of similar projects (count) and the scale of similar projects (scale).

## 5.2. Data Reliability Test

This study belongs to the case where two raters score the same test question or task, so the rating reliability estimation methods that can be used for this study include the Spearman correlation coefficient method, Kappa coefficient method, coefficient of contingency method, and Pearson product-moment correlation coefficient method. By a systematic comparison, the Spearman correlation coefficient method [69] was used to analyze the sample rate data, because this method has a wide range of applications, is suitable for situations where it is difficult to determine what distribution the two populations belong to, and the sample size is not limited. The significant correlations of the rating results for each project are all significant at a confidence level of 0.01, as shown in Table 15, which represents that the initial rating results from the eight respondents were valid. Additionally, during the interview process, the authors and each interviewee agreed on the time and location of the investigation in advance before conducting the one-to-one and face-to-face interviews. Furthermore, the authors recorded the investigation for verification after the consent of the interviewees was obtained.

**Table 15.** Significant correlation.

| Spearman's Rho | | A1 | A2 | Spearman's Rho | | B1 | B2 |
|---|---|---|---|---|---|---|---|
| **A1** | Correlation coefficient | 1.000 | 0.702** | **B1** | Correlation coefficient | 1.000 | 0.691** |
| | Sig. (twin tails) | . | 6.670E-12 | | Sig. (twin tails) | . | 1.772E-11 |
| | N | 72 | 72 | | N | 72 | 72 |
| **A2** | Correlation coefficient | 0.702** | 1.000 | **B2** | Correlation coefficient | 0.691** | 1.000 |
| | Sig. (twin tails) | 6.670E-12 | . | | Sig. (twin tails) | 1.772E-11 | . |
| | N | 72 | 72 | | N | 72 | 72 |
| Spearman's Rho | | C1 | C2 | Spearman's Rho | | D1 | D2 |
| **C1** | Correlation coefficient | 1.000 | 0.769** | **D1** | Correlation coefficient | 1.000 | 0.792** |
| | Sig. (twin tails) | . | 2.927E-15 | | Sig. (twin tails) | . | 1.263E-16 |
| | N | 72 | 72 | | N | 72 | 72 |
| **C2** | Correlation coefficient | 0.769** | 1.000 | **D2** | Correlation coefficient | 0.792** | 1.000 |
| | Sig. (twin tails) | 2.927E-15 | . | | Sig. (twin tails) | 1.263E-16 | . |
| | N | 72 | 72 | | N | 72 | 72 |

**. At a confidence level of 0.01, the correlation was significant.

*5.3. Evaluation Results*

5.3.1. Overview of the Evaluation Results

Figure 5 presents the evaluation results of four selected IC projects: Three projects (i.e., A, B, and C) had an optimal maturity and project D only reached an integration level in the "enablers" area; in the "results" area, all four projects had an integration level. Given the optimal level as the ideal level of IC maturity in real-life projects, the evaluation results indicate that the capability of implementing the IC projects in China still needs to be further enhanced, especially the "results" performance of the IC project construction. Specifically, the weakest aspect was found to be in R2 ("society"), where the three projects (i.e., B, C, and D) only reached the upgraded level, followed by R1 ("product") and R4 ("management and control"), where the four projects only reached the integrated level. Although a higher degree of maturity in the "enablers" area as a whole was deployed, there are still spaces for improvement, for example, all projects still need to be improved in E1 ("leadership"), where four projects are at the integrated level, and the capabilities of project C and D in E3 ("planning and control") and E4 ("technology and schema") are also at the integrated level and need to be improved to the optimal level. Overall, projects A and B outperformed the other two in the "results" area as well as their better organizational performance in the "enablers" area.

| | Project A | Project B | Project C | Project D |
|---|---|---|---|---|
| **E** | **81.1% Optimal level** | **80.8% Optimal level** | **76.8% Optimal level** | **74.7% Integrated level** |
| E1 | 62.5% Integrated level | 65.0% Integrated level | 58.8% Integrated level | 62.5% Integrated level |
| E2 | 94.0% Optimal level | 93.2% Optimal level | 91.1% Optimal level | 83.8% Optimal level |
| E3 | 78.5% Optimal level | 77.1% Optimal level | 69.9% Integrated level | 69.1% Integrated level |
| E4 | 78.8% Optimal level | 78.1% Optimal level | 74.8% Integrated level | 74.4% Integrated level |
| **R** | **64.8% Integrated level** | **57.3% Integrated level** | **52.8% Integrated level** | **51.3% Integrated level** |
| R1 | 62.7% Integrated level | 59.3% Integrated level | 57.0% Integrated level | 53.6% Integrated level |
| R2 | 68.8% Integrated level | 40.6% Upgraded level | 28.3% Upgraded level | 32.4% Upgraded level |
| R3 | 90.0% Optimal level | 95.0% Optimal level | 80.0% Optimal level | 80.0% Optimal level |
| R4 | 57.5% Integrated level | 60.0% Integrated level | 62.5% Integrated level | 58.8% Integrated level |

0~25% Initial level □    25%~50% Upgraded level □

50%~75% Integrated level □    75%~100% Optimal level □

**Figure 5.** Maturity level results of the selected IC projects.

### 5.3.2. Weak Areas of the Current IC Implementations

The target of IC project implementation is to achieve the "results" performance. At the criterion layer, the above maturity evaluation results indicated that weak areas in the "results" area mainly include "R1-product", "R2-society", and "R4-management and control"; and weak areas in the enablers" area include "E1-leadership", "E3-planning and control", and "E4-technology and schemas". According to the maturity-level description in each evaluation dimension of the ICMM, the performance improving the paths of the weak areas in both the "enablers" and "results" areas are discussed in Table 16. Additional details of the evaluation results are deployed as radar charts in Figures 6 and 7, which intuitively show the differences of the four selected projects at the evaluation indicator level, and the values from 1 to 5 in the figures refer to the average value of the actual rating score on each indicator of each project. We found that the performance of the four projects in the four "enablers" evaluation dimensions is not much different, but there is an obvious difference between the projects in the R2 dimension of the "results" area. According to Figures 6 and 7, the weak points of the four projects at the evaluation indicator level are further marked in Table 17.

**Table 16.** Improving path for weak areas of the selected projects.

| Weak Areas | | Project | Improving Path* | Improving Strategy Description |
|---|---|---|---|---|
| Enablers (How) | E1 | A, B, C, D | Integrated level→Optimal level | Project participants need to have a more clear and overall cognition of IC and enhance their willingness to implement IC. |
| | E3 | C, D | Integrated level→Optimal level | The project organization should make more detailed planning for progress prediction and resource applications and attempt to control the planning implementation by adopting the information tools (BIM, information software platform, etc.). |
| | E4 | C, D | Integrated level→Optimal level | The project organization should try to select more efficient and rationale technology and process schemas, so they need to enhance their capability to make decisions and adopt more innovative technologies. |
| Results (What) | R1 | A, B, C, D | Integrated level→Optimal level | The built entity adopting the IC technique system should be more functional and practical than that built by adopting the conventional method. |
| | R2 | A | Integrated level→Optimal level | The IC mode has brought the benefit of environment friendly and energy conservation. However, technology innovation needs to be further explored in the construction process for follow up applications. |
| | | B, C, D | Upgraded level→Optimal level | The IC mode has brought the basic benefit of environment friendly, more energy conservation and technology innovation need to be further explored in the construction process. |
| | R3 | A, B, C, D | Integrated level→Optimal level | The organizations have achieved the project objectives well, participants within the organizations can cooperate smoothly, and the members within the organizations can learn to grow. However, these temporary organizations need to try to establish a long-term cooperative relationship. |

*. It is assumed that the path of ascension is based on an optimal level of maturity.

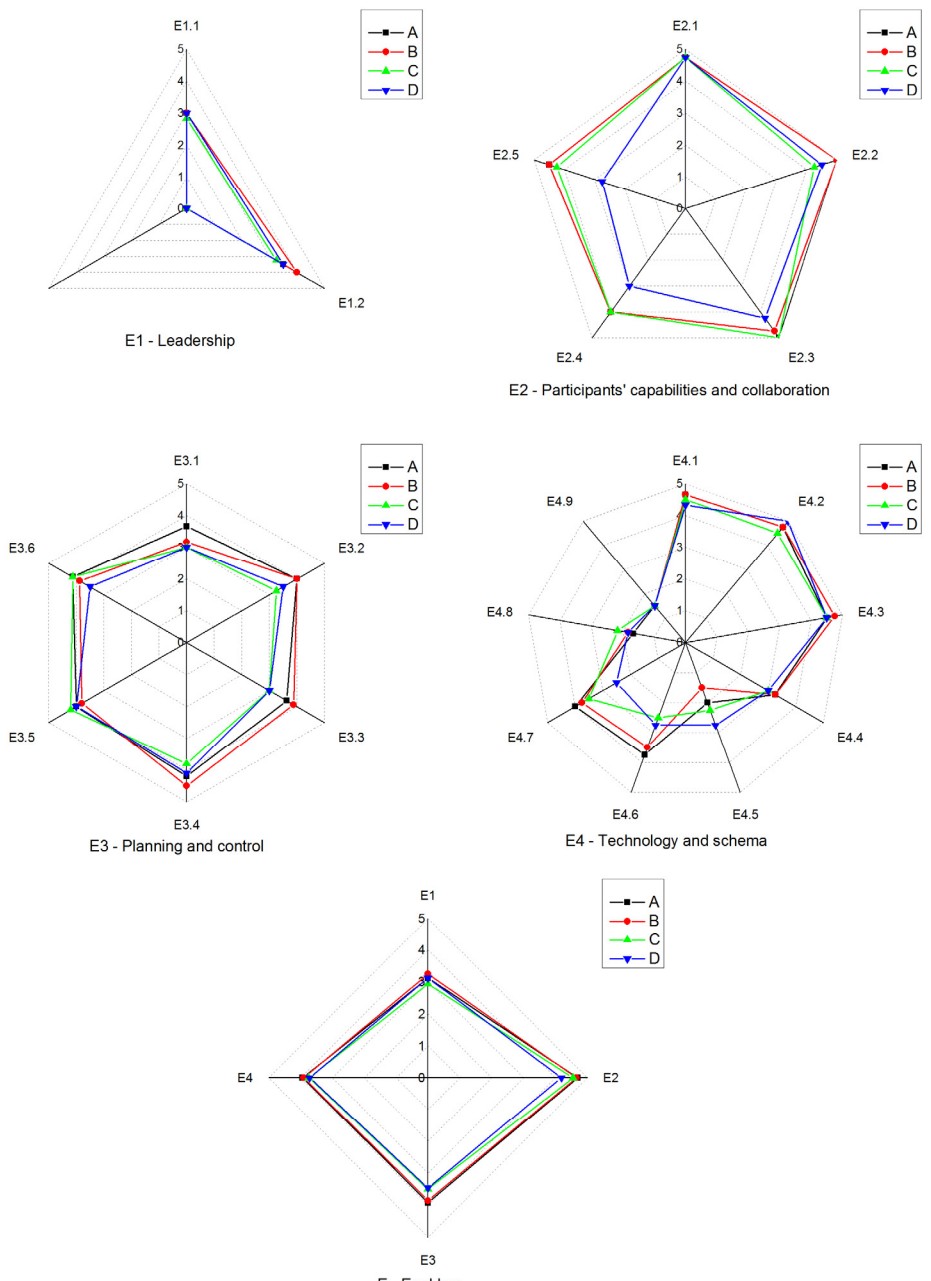

**Figure 6.** Rating results at the index layer of four selected projects in the "enablers" area.

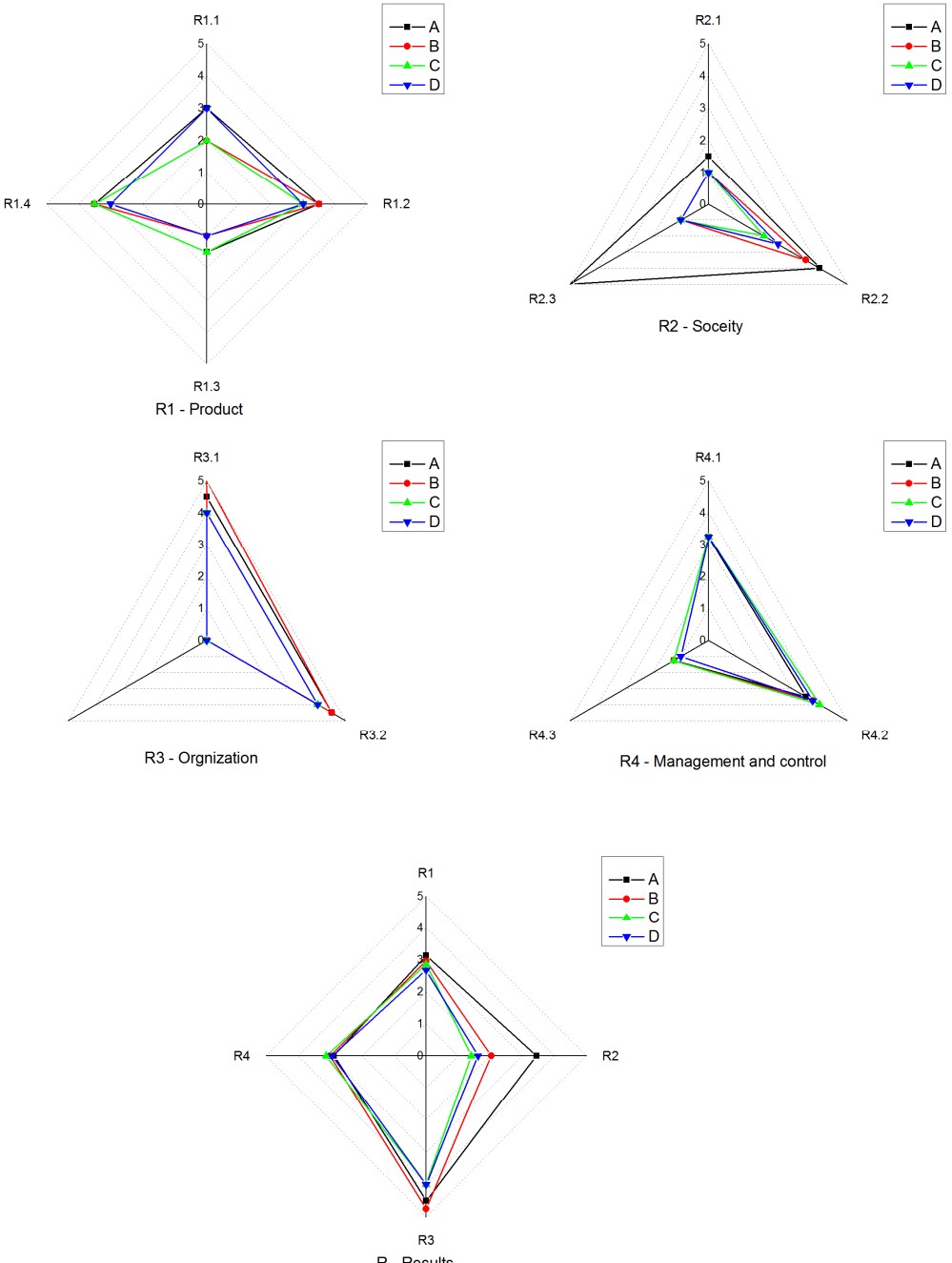

**Figure 7.** Rating score results at the index layer of four selected projects in the "results" area.

**Table 17.** Weak points at the index layer of the four selected projects.

| Dimension | Code | Indicators | A | B | C | D |
|---|---|---|:-:|:-:|:-:|:-:|
| Leadership (E1) | E1.1 | Owner's cognition and attitude | ● | ● | ● | ● |
| | E1.2 | Contractors' cognition and attitude | ● | | ● | ● |
| Participants' capabilities and collaboration (E2) | E2.1 | Designer's experience and ability | | | | |
| | E2.2 | Contractor's experience and ability | | | | |
| | E2.3 | Component supplier's experience and ability | | | | |
| | E2.4 | Cooperation willingness | | | | ● |
| | E2.5 | Collaboration channels | | | | ● |
| Planning and control (E3) | E3.1 | Goal setting | ● | ● | ● | ● |
| | E3.2 | Norms and standards | | | ● | ● |
| | E3.3 | Schedule control | ● | | ● | ● |
| | E3.4 | Change control | | | | |
| | E3.5 | Quality control | | | | |
| | E3.6 | Cost control | | | | ● |
| Technology and schema (E4) | E4.1 | Prefabricated technology system | | | | |
| | E4.2 | Advance work of IC design | | | | |
| | E4.3 | Design with component confirmation | | | | |
| | E4.4 | Detail design and process matching | ● | ● | ● | ● |
| | E4.5 | Component production | ◑ | ◑ | ◑ | ● |
| | E4.6 | Component transportation | ● | ● | ◑ | ● |
| | E4.7 | Construction with component assembly | | ● | ● | ◑ |
| | E4.8 | Industrialized decoration | ◑ | ◑ | ◑ | ◑ |
| | E4.9 | Operation preparation | ◑ | ◑ | ◑ | ◑ |
| Product (R1) | R1.1 | Prefabrication rate | ● | ◑ | ◑ | ● |
| | R1.2 | Practical performance | ● | ● | ● | ● |
| | R1.3 | Operating and maintenance savings | ◑ | ◕ | ◑ | ◕ |
| | R1.4 | Owner's satisfaction | ● | ● | ● | ● |
| Society (R2) | R2.1 | Technological innovation | ◑ | ◕ | ◕ | ◕ |
| | R2.2 | Environmentally friendly | | ● | ◑ | ◑ |
| | R2.3 | Honors or awards | | | ◕ | ◕ |
| Organization (R3) | R3.1 | Participants communication efficiency | | | | |
| | R3.2 | Participants' long-term cooperation willingness | | | | |
| Management and Control (R4) | R4.1 | Schedule | ● | ● | ● | ● |
| | R4.2 | Quality | ● | ● | | ● |
| | R4.3 | Cost | ◕ | ◕ | ◕ | ◕ |

◔. Score ≤ 1.25, initial level; ◑. 1.25 < Score ≤ 2.5; ●. 2.5 < Score ≤ 3.75; the blank space indicates that it has reached the desired (optimal) level.

## 6. Discussion

*6.1. Overall Maturity Analysis for the General Status of IC Projects in China*

According to the evaluation results in the multi-case study, a semi-structured interview was conducted. Eleven interviewees were asked to comment on the evaluation results at the criterion layer by referring to Figures 6 and 7 and Table 17. The commentary from the interview results was summarized as follows:

E1—"Leadership": The evaluation results show that the participants' cognition of IC is insufficient, and the active use of IC is also insufficiently motivated. The decision-makers of participants are inexperienced in IC implementation; there exist uncertainties about the IC implementation at this stage, as well as the economic benefits of the current IC implementation construction projects in China, which have not been obtained, so most of them are learning to explore the IC mode.

E2—"Participants' capabilities and collaboration": The evaluation results show that the participants' ability is enough to cope with the scope of duties and the depth of cooperation, and only the maturity level of the collaboration (i.e., E3.4 and E3.5) in project D needs to be further improved (in Table 17). The IC method requires the project-based organization to share information sufficiently and collaborate deeply, which is also the key problem in succeeding research and practices. To facilitate communication and collaboration between participants, the government has implemented a series of policies to encourage and reward the use of IC mode; communication efficiency is high in the internet age, and problems can be solved timely. All participants' cooperation matters are based on contracts, and the contract mechanism is mature and remains one of the most important ways to ensure good cooperation in temporary organizations.

E3—"Planning and control": The evaluation results show that some project organizations (in projects C and D) do not have enough capabilities of IC process planning and control. The management content is highly complicated, and the project progress and quality are required more strictly due to the increased number of management links in the IC mode, which involved more work in the planning and design stage, and mostly changed the work content on construction sites. However, traditional management methods by organizations are difficult to adapt to these changes. Additionally, the experts interviewed agreed that updating existing management methods and enhancing the abilities of managers is beneficial to process control.

E4—"Technology and schema": The evaluation results show that the criterion related to the component (i.e., E4.5-E4.9) has the lower maturity evaluation scores, which indicates that the aspects of component production, transportation, construction with component assembly, industrial decoration, and operation preparation are still in a developing stage and facing numerous technical challenges (e.g., structural stability, firmness of the joint) and numerous risks (e.g., the safety of workers, loss of components, and structural integrity) during the practical implementation process. Additionally, the uncertainties of the onsite working environment and organizations' insufficient experience are also the main reasons.

R1—"Product": The evaluation results show that the IC mode has brought meager benefits in the functionality and practicality of the building entity. Because the traditional delivery methods (e.g., DBB or GC) are still used, the contractor only needs to make a profit by controlling the cost as much as possible during the construction process while meeting the requirements from the owner and designers. Given the adoption of new construction processes under the IC mode, new technical and management problems would influence the performance of construction products.

R2—"Society": The evaluation results show that the most remarkable social benefit achieved is environmentally friendly. Conversely, these projects have barely gained energy conservation and a social reputation, which represents the comprehensive performance of an IC project, yet the satisfactory breakthrough of technology innovation has not been made. According to the results of the interview, the existing projects mainly adopt existing mature and stable IC technologies, for example, BIM technology

and information platform technologies are also developing gradually and are not mature enough to be adopted popularly.

R3—"Organization": The evaluation results show that all the sample projects have achieved the optimal level of organizational performance, which is in accordance with the evaluation results of E2-"participants' capabilities and collaboration". Overall, the organization in the current IC projects, limited to existing cooperative delivery models (DBB or GC), has largely maximized benefits in terms of cooperation. It has been manifested that participants have a strong willingness to cooperate and the diversification of cooperation channels is selected. In the aspect of long-term cooperation, interviewees also demonstrated that long-term cooperation is an efficient cooperation scheme that can reduce the communication and cooperation cost between participants. Additionally, this new construction mode (IC) provides a driving force for organizational members to learn, which also contributes to the high degree of cooperation satisfaction of the participants.

R4—"Management and control": The evaluation results show that the sample projects gain slight advantages over the traditional construction mode in terms of "schedule" and "quality" but not in terms of "cost". The interview results combining the comparative analysis between the process control items in the "planning and control" and the developing state of "technology and schema" evaluation dimensions show that the costs are largely higher due to the need to improve existing technology and the management process of IC mode.

## 6.2. Model Evaluation

The proposed ICMM has high reliability and applicability, according to the evaluation conclusion from the case study discussion and the direct results of the face-to-face semi-structured interviews. On the one hand, the interviewees gave the selected project an evaluation score that passes the test of inter-rater reliability, which shows that the data sources are reliable. On the other hand, the interview results of evaluating the ICMM are presented as follows.

(1) Indicators. Most experts agreed that the eight evaluation dimensions are sufficient to cover the key aspects of IC projects in the "enables" and "results" areas. A small number of experts suggested that more criteria may be needed, e.g., "incentives by the government". The majority view is that the proposed model provides a generic template. Specific projects located in different countries or regions depending on situations of market-driven or government intervention can customize their models by adding additional criteria or removing existing criteria.

According to the experts, some dependencies exist between the eight categories of relationship indicators, e.g., participants' attitudes towards IC (E1-"leadership") influence willingness to cooperate. However, they believed that eight key relationship indicators should be treated separately because each of them represents the main aspect of IC projects.

All the experts agreed that no clear omissions existed at the index level. At the same time, the experts believed that the indicators were put in the appropriate categories.

(2) Weights. Indicators of the two evaluation areas were assigned weights by the AHP approach. Experts believe that the assignment results are generally in line with the practical situation and explain the different weights of some indicators. The weight of "honors and awards" is higher than the weight of "schedule" and "cost", for example, because they believe that the acquisition of "honors and awards" is more beneficial to the sustainable development of construction team and the following application and operation of the project, while the "cost" and "schedule" are only the short-term goals of the construction stage.

(3) Maturity level. All experts approved of describing IC projects at four maturity levels. Following the interviewer's suggestion, some experts positioned their existing IC projects. An important finding is that the greater the experience of participating enterprises in IC projects, the greater the maturity of the projects in "enablers" aspects; the "results" performance, where the factors affecting are complex, do not have the same characteristics with the "enablers" aspects, weaknesses of

the organizational management in IC projects, and strategies for improving project performance can be identified from the corresponding "enablers" areas.

(4) Implementation process. All experts considered that the evaluation procedure is well developed systematically. Most experts approved that the complete procedure can be simplified by merging the questionnaire survey with the interviews.

(5) Practical implications. All experts supported that the proposed model can be used not only for public building projects but also for residential projects by IC mode. Additionally, a high degree of consensus existed between the experts that the proposed model can be used to reflect the performance status of IC projects and provide improvement strategies from a governance perspective very well. Yet, the application of the ICMM also needs to obey several essential principles. The presented ICMM is only applicable to Chinese scenarios because the delivery modes (e.g., DBB, GC), the contract and bidding laws, and the organizational culture may restrict cross-organizational application. The evaluation by the ICMM is limited to the project level, rather than to the enterprise or industry levels. In the spatial perspective, a single project level does not involve the problems of the regional organization network and repetitive cooperation, which the ICMM does not involve. In the time perspective, the ICMM can only be applied during the construction cycle of a project (not including the maintenance and removal phase after completion).

In all, the ICMM can fully reflect the IC maturity level of the prefabricated construction project. All the experts approved of describing IC at four maturity levels. The indicators and evaluation items can be understood by interviewees, and the scores can provide correct feedback. Besides, the conducted multi-case study proved that the evaluation results in the above four cases can help project managers rate the project status and provide improvement paths from the perspective of governance. Evaluating IC maturity in other national or regional contexts may determine more appropriate indicator systems following the presented process of ICMM development in this study, which is also one of the potential research directions.

## 7. Conclusions

The evaluation of IC maturity for building projects in China is necessary. Such an evaluation can help IC method applicators obtain a clear view of the status quo of IC to clarify numerous fuzzy challenges and improve weak areas. This study established an ICMM for the project's IC maturity with two evaluating areas ("enablers" and "results") based on the framework of the EFQM excellence model, CMM theory, existing IC evaluation research, and practical characteristics of China's IC projects. To establish the ICMM, the evaluation indicator system was initially identified. The "enablers" area involves 4 evaluation dimensions, 22 indicators, and 57 evaluation items. The "results" area involves 4 evaluation dimensions, 12 indicators, and 15 evaluation items; the evaluation model set four maturity rating levels, namely, initial, upgraded, integrated, and optimal levels. To validate the ICMM, a multi-case study with the selection of four typical IC projects in Shanghai, structural interviews and discussions on the reliability and applicability of the ICMM were conducted. The sample data obtained from the survey successfully passed the test of inter-rater reliability. The IC maturity evaluation results reflect the maturity level of the selected projects in terms of the "enablers" and "results", respectively. All respondents in the survey study approved the evaluation results and agreed that the evaluation results by ICMM can help managers to clarify weak areas in the current project and provide improvement paths in the perspective of governance, and the applicability and operability of the constructed model were demonstrated via the above case analysis and model evaluation by expert interviews.

Key findings of the IC applications in China were found as follows:

- The government's leadership has a strong impact on the attitude of owners and participants, but the cognition of IC and leadership of the participants in IC projects need to be improved (E1).

- Cooperation between participants within a project organization is more adequate than the cross-organization cooperation under the current delivery method and social environment (E2, R3).
- The current management and control process need to be optimized, and the IC mode has no advantage over the traditional mode in cost (E3, R4).
- Compared with the traditional construction mode, the built products have little difference in practical function and the satisfaction of owners, but they have obvious advantages in the environment friendly aspect, which are beneficial to the sustainability of the construction industry (R1, R2).
- The current IC mode still needs to be improved, including many technical problems in the links of design, component supply, and site assembly, which are still developing generally (E4).

Although the objectives of this study were achieved, several limitations to the conclusions exist, which can be drawn from the results. First, this study was limited to the selection of four typical building projects for the proposed ICMM validation. Due to the small sample size of the project, the relationship between project attributes and the maturity of IC projects could not be quantitatively explored. Additionally, in the analysis of model indicators, the internal relationship between model indicators and the sensitivity of indicators to project attributes could not be quantitatively verified. This study analyzed the reliability and applicability of ICMM only from a qualitative perspective. Second, maturity enhancement is not a static process but rather a dynamic and continuous one, of which this study is limited to considering revealing the IC maturity of projects in only one construction cycle.

Therefore, by combining the basic theoretical contributions and limitations of this study, future research should focus on exploring the relationship between project attributes and the maturity of IC projects, and the correlation between indicators of the ICMM and the sensitivity of indicators to project attributes by using a larger size of sample data from the perspective of quantitative analysis. Additionally, the evolution path of ICMM in prefabricated construction building projects could be studied over a longer period from a dynamic perspective.

**Supplementary Materials:** The following are available online at http://www.mdpi.com/2071-1050/12/10/4029/s1, a pdf file: Questionnaire for Industrialized Construction Maturity Evaluation of Building Projects.

**Author Contributions:** G.W. and H.L. (Huan Liu) came up with this research idea, conducted the literature review, performed the theoretical framework model and wrote the original draft. J.L. contributed to data collection and participated in the data analysis. H.L. (Heng Li) and X.L. provided some comments and supplemented the research framework. All authors have read and agreed to the published version of the manuscript.

**Funding:** This research received no external funding.

**Acknowledgments:** This research is funded by the National Natural Science Foundation of China (Grant No. 71771172 and 71471138), the Shanghai Municipal Science and Technology Commission (Project code: 17DZ1203602) in Mainland China and the Research Institute for Sustainable Urban Development (RISUD) in Hong Kong (Project code: 5-ZJLG). The authors are grateful to respondents during the data collection processes. The authors also would like to thank the editor and the reviewers for their helpful suggestions.

**Conflicts of Interest:** The authors declare no conflict of interest.

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
