# Peer review of "A Building Project-Based Industrialized Construction Maturity Model Involving Organizational Enablers: A Multi-Case Study in China"

_sustainability, doi:10.3390/su12104029_

Round 1
Reviewer 1 Report
Dear Authors,
This is a very interesting area of research i.e. the Industrialized Construction. The contribution of this article to the Body of Knowledge is the hybrid model that the authors propose concerning a new evaluation system that combines Capability Maturity Model (CMM) and EFQM in the area under investigation. To do so, an AHP was used to prioritize the indicators after scrutinizing the literature for their identification. Then the validation process was performed by testing the model in 4 building projects in China.
Please find in the following section my suggestions for further improvement of your work.
- Moderate English changes are required so I would suggest a word proofreading of the whole text.
- There should be more explanation about the results of figures 5 and 6, the rating is from 1 to 5, what does this rating indicate? There is no reference in the text.
- I would expect a sensitivity analysis of the indicators on the whole performance of the process.
- The validation of the model is limited in the building sector, what about the other sectors in construction, i.e. infrastructure projects. There no reference in the paper concerning this matter.
Author Response
Response to Reviewer 1 Comments
Point 1: Moderate English changes are required so I would suggest a word proofreading of the whole text.
Response 1: A careful English, grammar and syntax revision of the entire manuscript has been performed. Thanks for mentioning us to improve the language of the revised manuscript, and we hope that our revised manuscript addresses any similar concern.
Point 2: There should be more explanation about the results of figures 5 and 6, the rating is from 1 to 5, what does this rating indicate? There is no reference in the text.
Response 2: We [the authors] thank the reviewer for these valuable suggestions to improve our manuscript. The authors have added the text of the explanation of the figures 6 and 7 (in the revised manuscript) and explained what the rating from 1 to 5 indicate in the revised manuscript. It has been presented as follows:
“Additional details of the evaluation results are deployed as radar charts in Figures 6 and 7, which intuitively show the differences of four selected projects at the evaluation indicator level, and the values from 1 to 5 in figures refer to the average value of the actual rating score on each indicator of each project. We found that the performance of the four projects in the four evaluation dimensions of the “Enablers” area is not much different, but there is an obvious difference in the R2 dimension of the “Results” area. According to Figures 6 and 7, the weak points of the four projects at the evaluation indicator level were further marked in Table 17.” Please see lines 452 to 458 in the revised manuscript.
Point 3: I would expect a sensitivity analysis of the indicators on the whole performance of the process.
Response 3: We [the authors] thank the reviewer for these valuable suggestions. Sensitivity analysis may be a good method to help this study build a more reasonable and reliable evaluation indicator system. However, this study adopts a multi-case empirical research method, that is, the sample size of the building project for IC maturity evaluation is not enough to support the sensitivity analysis of indicators. This paper only discusses and evaluates the evaluation indicators of ICMM qualitatively through expert interviews. Besides, we also mentioned the problem of the small sample size of projects in the research limitations of this paper, and that the follow-up research needs to expand the empirical sample size to explore the relationship between evaluation dimensions or indicators. It has been presented as follows:
Limitations: “Although the objectives of this study are achieved, several limitations to the conclusions exist, which can be drawn from the results. First, this study was limited to select four typical building projects for the proposed ICMM validation. Due to the small sample size of the project, the relationship between project attributes and the maturity of IC projects cannot be quantitatively explored. Also in the analysis of model indicators, the internal relationship between model indicators and the sensitivity of indicators to project attributes cannot be quantitatively verified. This study analyzed the reliability and applicability of ICMM only from a qualitative perspective.” Please see lines 624 to 630 in the revised manuscript.
Future study: “Therefore, by combining the basic theoretical contributions and limitations of this study, future research should focus on exploring the relationship between project attributes and the maturity of IC projects, the correlation between indicators of the ICMM and the sensitivity of indicators to project attributes by using a larger size of sample data from the perspective of quantitative analysis.” Please see lines 633 to 636 in the revised manuscript.
Point 4: The validation of the model is limited in the building sector, what about the other sectors in construction, i.e. infrastructure projects. There no reference in the paper concerning this matter.
Response 4: We [the authors] thank the reviewer for inspirational suggestions and comments. Based on the context of China's construction industry, the current industrialized construction policy in China mainly aimed at building projects, and there are many technical and management difficulties in adopting the transformative industrialized construction mode that needs to be alleviated by reasonable and effective evaluation methods. However, whether and how to implement the industrialized construction mode in the construction of large-scale public infrastructure (such as roads, bridges, airports, etc.) is not within the scope of this study, but it is a subject worth exploring in the follow-up study. The authors have added the word “Building” into the title and revised the context of China's construction industry in the revised manuscript, it has been presented as follows:
“A Building Project-based Industrialized Construction Maturity Model Involving Organizational Enablers: A Multi-case Study in China”. Please see lines 2 to 4 in the revised manuscript.
“Considering the advantages of the IC mode, cities such as Beijing, Shanghai, and Shenzhen in China have carried out a series of IC technology experiments on prefabricated construction building projects and implemented preferential policies and incentives for builders. However, the desired efficiency (e.g., cost savings and duration shortening) has yet been achieved. Such problems related to market demand, production networks, technology docking, and supply consolidation has emerged in the low productivity of large-scale onsite construction and prevented the progress of industrialization of construction [1].” Please see lines 42 to 49 in the revised manuscript.
Reference (1):
- Zhang, J.; Long, Y.; Lv, S.; Xiang, Y. BIM-enabled Modular and Industrialized Construction in China. In Proceedings of the Procedia Engineering; 2016; pp. 145, 1456–1461.

Reviewer 2 Report
The article provides a model for evaluating the maturity of industrialized construction projects. This model has involved organizational enablers by employing the framework of the EFQM Excellence. The model has been tested on a multi-case study and its result can help managers identify the weak areas of the current industrialized construction project and performance improvement path from the organizational perspective.
The article is well structured.
The main comment is that the article does not give any meaning to how the proposed model is related to the challenges and opportunities to support sustainable construction (the article applies to prefabricated construction projects).
Author Response
Response to Reviewer 2 Comments
Point 1: The main comment is that the article does not give any meaning to how the proposed model is related to the challenges and opportunities to support sustainable construction (the article applies to prefabricated construction projects).
Response 1: We [the authors] thank the reviewer for these valuable suggestions/comments to improve our manuscript. The industrialized construction has been given increasing attention due to the challenges that traditional construction methods encounter. Therefore, developing a model to evaluate the maturity of an industrialized construction project is worthwhile to study and relevant to the Sustainability journal. The authors have also highlighted the signification of implement IC mode to sustainability in the revised manuscript. It has been presented as follows:
“Industrialized construction (IC), as a promising construction mode that can enhance production efficiency and reduce the labor intensiveness in the construction industry [1–5], has played an important role in facilitating the sustainability performance of construction projects recently [6].” Please see lines 35 to 37 in the revised manuscript.
Reference (2):
- Jaillon, L.; Poon, C.S. The evolution of prefabricated residential building systems in Hong Kong: A review of the public and the private sector. Automation in Construction 2009, 18(3), 239–248.
- Jaillon, L.; Poon, C.S. Life cycle design and prefabrication in buildings: A review and case studies in Hong Kong. Automation in Construction 2014, 39, 195–202.
- Johnsson, H.; Meiling, J.H. Defects in offsite construction: Timber module prefabrication. Construction Management and Economics 2009, 27(7), 667–681.
- McGraw Hill Construction Prefabrication and Modularization: increasing productivity in the construction industry; 2011;
- Taylor, T.R.B.; Ford, D.N.; Reinschmidt, K.F. Impact of Public Policy and Societal Risk Perception on U.S. Civilian Nuclear Power Plant Construction. Journal of Construction Engineering and Management 2012, 138(8), 972–981.
- Shen, L.Y.; Li Hao, J.; Tam, V.W.Y.; Yao, H. A checklist for assessing sustainability performance of construction projects. Journal of Civil Engineering and Management 2007.

Reviewer 3 Report
The authors developed a model that evaluates the maturity of industrialized construction projects by integrating the Capability Maturity Model (CMM) and the European Foundation for Quality Management (EFQM) excellence model. The industrialized construction has been given increasing attention due to the challenges that traditional construction methods encounter. Therefore, developing a model to evaluate the maturity of an industrialized construction project is worthwhile to study and relevant to the Sustainability journal. The followings are the reviewer's comments and suggestions to improve the quality of the current manuscript.
- The literature review section could be improved. The ICMM builds upon the CMM and EFQM excellence model as the authors mentioned. However, the authors did not provide enough introduction to the CMM and EFQM excellence model. The authors may want to include a more detailed explanation of the CMM and EFQM excellence model in the literature review. For example, the authors should include dimensions of enablers and results when the authors introduced the EFQM excellence model. Although readers may be able to identify the dimensions after reading the model development section, a comprehensive introduction to the EFQM excellence model should be provided in the literature review section.
- The explanation of indicator identification also could be improved. The authors may want to provide the definition of each indicator and justification of including the indicator. More detailed explanations of indicators may improve the internal validity of the model.
- The reviewer is not clear about the AHP process. There are two levels (i.e., dimension and indicators) in the model. Is the weight of each indicator determined based on the comparison with indicators in the same dimension after determining the weight of each dimension separately? In other words, the AHP process first determines 1) weights of E1, E2, E3, and E4 2) determines the relative weight of E1.1 and E1.2. Then, the final weight of E1.2 will be the weight of E1 multiplied by the relative weight of E1.2. Is the weight of each indicator determined by comparison with all other indicators at the same time? Please elaborate on the AHP process with two levels.
- The author may want to provide an additional explanation of the evaluation of each indicator to determine the maturity of the project. For example, the reviewer is not sure how the participant evaluates each indicator on the Likert 5 scale. It would be better if the authors include items in the questionnaire to evaluate the projects.
- The reviewer is not sure about the evaluating process in the case study. Did the participants evaluate all the four projects? If then, it would be very challenging to evaluate projects that a participant was not involved because he/she did not have enough information to evaluate the project.
- Line 367: Please explain the differences between the DBB and delivery method of the General-contractor. In the reviewer's understanding, DBB hires a general contractor to execute the construction process.
- The authors provided a significant correlation to support the validity of the model. Please provide a justification that supports the use of correlation. The intraclass correlation (ICC) would be better to show the reliability of the evaluation method.
Author Response
Response to Reviewer 3 Comments
Point 1: The literature review section could be improved. The ICMM builds upon the CMM and EFQM excellence model as the authors mentioned. However, the authors did not provide enough introduction to the CMM and EFQM excellence model. The authors may want to include a more detailed explanation of the CMM and EFQM excellence model in the literature review. For example, the authors should include dimensions of enablers and results when the authors introduced the EFQM excellence model. Although readers may be able to identify the dimensions after reading the model development section, a comprehensive introduction to the EFQM excellence model should be provided in the literature review section.
Response 1: We [the authors] thank the reviewer for these valuable suggestions/comments to improve our manuscript. Following the above constructive suggestions, the authors have added a more detailed introduction of the CMM and the EFQM excellence model into the literature review section in the revised manuscript. It has been presented as follows:
“The CMM is organized into five maturity levels (i.e., initial, repeatable, defined, managed, and optimizing). Except for Level 1, each maturity level is decomposed into several key process areas that indicate the areas an organization should focus on to improve its software process [1]. The rating components of the CMM, to assess an organization’s process maturity, are its maturity levels, key process areas, and their goals. Each key process area is further described by informative components: key practices, sub practices, and examples. The key practices describe the infrastructure and activities that contribute most to the effective implementation and institutionalization of the key process area.” Please see lines 129 to 136 in the revised manuscript.
“The EFQM Excellence Model consists of the “Enablers” area with five dimensions (i.e., “Leadership”, “People”, “Strategy”, “Partnerships & Resources”, and “Processes, Products & Services”) and the “Results” area with four dimensions (i.e., “People Results”, “Customer Results”, “Society Results”, and “Business Results”). “Enablers” describe what an organization should do and how to achieve its organizational goals. “Results” focus on what is important to the key stakeholders.” Please see lines 159 to 164 in the revised manuscript.
Point 2: The explanation of indicator identification also could be improved. The authors may want to provide the definition of each indicator and justification of including the indicator. More detailed explanations of indicators may improve the internal validity of the model.
Response 2: We [the authors] thank the reviewer for these valuable suggestions/comments to improve our manuscript. The authors have added more details of justification of including each indicator in Table 5 and Table 6 in the revised manuscript. It has been presented as follows:
“Table 5 and Table 6 present the indicators in the “Enablers” area and the “Results” area, respectively.” Please see lines 263 to 271 and Tables 5 and 6 in the revised manuscript.
Point 3: The reviewer is not clear about the AHP process. There are two levels (i.e., dimension and indicators) in the model. Is the weight of each indicator determined based on the comparison with indicators in the same dimension after determining the weight of each dimension separately? In other words, the AHP process first determines 1) weights of E1, E2, E3, and E4 2) determines the relative weight of E1.1 and E1.2. Then, the final weight of E1.2 will be the weight of E1 multiplied by the relative weight of E1.2. Is the weight of each indicator determined by comparison with all other indicators at the same time? Please elaborate on the AHP process with two levels.
Response 3: We [the authors] thank the reviewer for these valuable suggestions/comments to improve our manuscript. The authors agreed with the reviewer and added a more detailed description of the calculation process and a separate graph to show the weighting calculation process and results (%). It has been presented as follows:
“Step 5: The weights are calculated by completing Steps 1 to 4 for all levels in the hierarchy (i.e., dimensions and indicators) of the ICMM. The weight of each indicator is determined based on the comparison with indicators in the same dimension after determining the weight of each dimension separately. Specifically, in the “Enablers” area, weights of evaluation dimensions Pi (i = E1, E2, E3, and E4) and the relative weights of indicators to their corresponding evaluation dimension Rij (e.g., R12 means the relative weight of E1.2 to E1) are first determined separately. Then, the final weight of the indicator Pij will be the weight of its corresponding dimension Pi multiplied by the relative weight of the indicator Rij. The weighting calculation rules and results (%) of the indicators are presented in Figure 3.” Please see lines 315 to 323 and Figure 3 in the revised manuscript.
Point 4: The author may want to provide an additional explanation of the evaluation of each indicator to determine the maturity of the project. For example, the reviewer is not sure how the participant evaluates each indicator on the Likert 5 scale. It would be better if the authors include items in the questionnaire to evaluate the projects.
Response 4: We [the authors] thank the reviewer for these valuable suggestions/comments to improve our manuscript. The authors have attached the questionnaire to the text through a public URL link, and readers can directly click to obtain the detailed content of the questionnaire. The questionnaire contains all evaluation items that correspond to the evaluation indicators for rating the actual projects. It has been presented as follows:
“During the rating process, all items of the scale in a questionnaire are measured using the Likert 5-point scale. This scaling approach is concise and easy to answer and is widely used in tools to measure respondents’ views, beliefs, and attitudes. The degree of conformity between the description and the actual situation in the rating questionnaire was carried out by the five levels of "extremely conformity ", "reluctantly conformity", "uncertain", "unconformity" and "extremely non-conformity". The questionnaire can be accessed at https://huanhannah.coding.net/s/397dd461-12a9-4645-a96e-11c206e2ed5b.” Please see lines 274 to 280 in the revised manuscript.
Point 5: The reviewer is not sure about the evaluating process in the case study. Did the participants evaluate all the four projects? If then, it would be very challenging to evaluate projects that a participant was not involved because he/she did not have enough information to evaluate the project.
Response 5: We [the authors] thank the reviewer for these valuable suggestions/comments to improve our manuscript.
It is one of the qualifications for using the ICMM to assess the sample project’s IC maturity that raters who must understand the objective situation of the projects comprehensively. The raters scores according to the actual situation of the project, because it is not correct to find a person who does not understand the actual situation of the project to comprehensively evaluate the project. The case study part of this study is mainly to present the application effect of ICMM and have a qualitative analysis of ICMM. The results of the case study show that the model achieves the application goals, that is, to reveal the weaknesses of the project and provide improvement suggestions for the project organization from the perspective of project governance. The corresponding text in the “Research Methods” Section has been revised. It has been presented as follows:
“This study has invited ten managers (construction practitioners) from these four selected projects to participate in the model evaluation interview and two senior engineers in each project are selected to participate in rating the corresponding project that they involved in, the specific information of the survey participants is presented in Section 5.” Please see lines 232 to 236 in the revised manuscript.
Point 6: Line 367: Please explain the differences between the DBB and delivery method of the General-contractor. In the reviewer's understanding, DBB hires a general contractor to execute the construction process.
Response 6: We [the authors] thank the reviewer for these valuable suggestions/comments to improve our manuscript. In this study, the detailed meanings of delivery methods of DB, DBB and CM-at GC have referred to the cited publication, and the authors have revised the corresponding text in the revised manuscript. It has been presented as follows:
“in terms of project delivery method, projects A and D adopt design-building (DB) and design-bid-building (DBB) respectively, projects B and C adopt the method of construction management at general contractor (CM-at GC) [2].” Please see lines 388 to 391 and Table 13 in the revised manuscript.
Point 7: The authors provided a significant correlation to support the validity of the model. Please provide a justification that supports the use of correlation. The intraclass correlation (ICC) would be better to show the reliability of the evaluation method.
Response 7: We [the authors] thank the reviewer for these valuable suggestions/comments to improve our manuscript.
The data distribution of this case study does not meet the ICC calculation conditions. As mentioned in point 5, we select project practitioners who have a good understanding of the whole process of the sample project to rate the project separately. For example, A-1 and A-2 only evaluate project A and do not participate in the B, C or D projects. This study belongs to the case where two raters score the same test question or task, so the rating reliability estimation methods that can be used for this study include the Spearman correlation coefficient method, Kappa coefficient method, Coefficient of contingency method and Pearson product-moment correlation coefficient method. We chose the Spearman correlation coefficient method is because the method has a wide range of applications, is suitable for situations where it is difficult to determine what distribution the two populations belong to, and the sample size is not limited.
Therefore, the authors have added the reason for using the Spearman correlation coefficient method for rating reliability tests and revised the table with the corresponding text in the revised manuscript. It has been presented as follows:
“This study belongs to the case where two raters score the same test question or task, so the rating reliability estimation methods that can be used for this study include the Spearman correlation coefficient method, Kappa coefficient method, Coefficient of contingency method and Pearson product-moment correlation coefficient method. By a systematic comparison, the Spearman correlation coefficient method [3] was used to analyze the sample rate data, because this method has a wide range of applications, is suitable for situations where it is difficult to determine what distribution the two populations belong to, and the sample size is not limited. The significant correlations of the rating results for each project are all significant at a confidence level of 0.01, as shown in Table 15, which represents that the initial rating results from the eight respondents are valid.” Please see lines 411 to 420 and Table 15 in the revised manuscript.
Reference (1):
- Paulk, M. Capability Maturity Model for Software. In Encyclopedia of Software Engineering; 2002.
- Huimin, L.; Zhuofu, W. Grey relational grade decision model for selection of project delivery system. In Proceedings of the 2009 IEEE International Conference on Grey Systems and Intelligent Services (GSIS 2009); IEEE, 2009; pp. 1033–1037.
- Cleff, T. Exploratory data analysis in business and economics: An introduction using spss, stata, and excel; 2014; ISBN 9783319015170.

Round 2
Reviewer 1 Report
Thank you. You have a answer sufficiently my points.